# The Mineral Aerosol Profiling from Infrared Radiances (MAPIR) algorithm: version 4.1 description and evaluation

Sieglinde Callewaert[1], Sophie Vandenbussche[1], Nicolas Kumps[1], Arve Kylling[2], Xiaoxia Shang[3], Mika Komppula[3], Philippe Goloub[4], and Martine De Mazière[1]

[1]Royal Belgian Institute for Space Aeronomy (BIRA-IASB), Avenue Circulaire 3, 1180 Brussels, Belgium
[2]Norwegian Institute for Air Research (NILU), P. O. Box 100, 2027 Kjeller, Norway
[3]Finnish Meteorological Institute (FMI), P.O. Box 1627, 70211, Kuopio, Finland
[4]Laboratoire d'Optique Amosphérique (LOA), Université des Sciences et Technologies de Lille, Villeneuve d'Ascq, France

**Correspondence:** Sieglinde Callewaert (sieglinde.callewaert@aeronomie.be), Sophie Vandenbussche (sophie.vandenbussche@aeronomie.be)

**Abstract.** The Mineral Aerosol Profiling from Infrared Radiances (MAPIR) algorithm retrieves vertical dust concentration profiles from cloud-free IASI thermal infrared (TIR) radiances using the Rodgers Optimal Estimation Method (OEM). We describe the new version 4.1 and evaluation results. Main differences with respect to previous versions are the Levenberg-Marquardt modification of the OEM, the use of the logarithm of the concentration in the retrieval and the use of RTTOV for in-line radiative transfer calculations. The dust aerosol concentrations are retrieved in seven 1 km thick layers centered at 0.5 to 6.5 km. A global data set of the daily dust distribution was generated with MAPIR v4.1 covering September 2007 to June 2018, with further extensions planned every six months. The post-retrieval quality filters reject about 16 % of the retrievals, a huge improvement with respect to the previous versions where up to 40 % of the retrievals were of bad quality. The median difference between the observed and fitted spectra of the good quality retrievals is 0.32 K, with lower values over oceans. The information content of the retrieved profiles shows dependency on the total aerosol load due to the assumption of a log-normal state vector. The median degrees of freedom in dusty scenes (min 10 μm AOD of 0.5) is 1.4. An evaluation of the aerosol optical depth (AOD) obtained from the integrated MAPIR v4.1 profiles was performed against 72 AERONET stations. The MAPIR AOD correlates well with the ground-based data with a mean correlation coefficient of 0.66 and values as high as 0.88. Overall, there is a mean AOD (550 nm) positive bias of only 0.04 with respect to AERONET, which is an extremely good result. The previous versions of MAPIR were known to largely overestimate AOD (about 0.28 for v3). A second evaluation exercise was performed comparing the mean aerosol layer altitude from MAPIR with the mean dust altitude from CALIOP. A small underestimation was found, with a mean difference of about 350 m (standard deviation of about 1 km) with respect to the CALIOP cumulative extinction altitude, which is again considered very good as the vertical resolution of MAPIR is 1 km. In the comparisons against AERONET and CALIOP, a dependency of MAPIR on the quality of the temperature profiles used in the retrieval is observed. Finally, a qualitative comparison of dust aerosol concentration profiles was done against lidar measurements from two ground-based stations (M'Bour and Al Dhaid) and from the CATS instrument onboard the ISS. MAPIR v4.1 showed the ability to detect dust plumes at the same time and with a similar extent as the lidar instruments. This new MAPIR version shows a great improvement of the accuracy of the aerosol profile retrievals with respect to previous

versions, especially so for the integrated AOD. It now offers a unique 3-D dust data set, which can be used to gain more insight in the transport and emission processes of mineral dust aerosols.

# 1 Introduction

Aerosols are solid or liquid particles such as desert dust, sea salt, volcanic ash, sulfate, black carbon and particulate organic matter which are suspended in Earth's atmosphere. Of all aerosol types, windblown mineral dust is the one with the highest mass burden, originating from soils in arid and semi-arid regions. These small particles can be transported over large distances to be finally deposited back on the surface of the Earth (Knippertz and Stuut, 2014).

The presence of mineral dust in the atmosphere has consequences for a wide range of aspects of life on Earth as it can cause respiratory diseases, reduce visibility and act as a fertilizer both on ocean and land. But it can also alter the radiative budget, have an impact on cloud microphysics, weather and climate dynamics and atmospheric chemistry (Knippertz and Stuut, 2014). Dust particles alter the radiative budget of the Earth through the aerosol direct and indirect effect. The direct effect is caused by the thermal emission of the dust particles and most importantly by the absorption and scattering of the solar shortwave and thermal longwave radiation by these particles. Dust aerosols can also act as cloud condensation nuclei and alter the lifetime and properties of clouds, thereby influencing the hydrological cycle and having an indirect effect on the radiation budget of the Earth (Boucher et al., 2013). Moreover, mineral dust affects the temperature profiles in the troposphere, which may impact the general atmospheric stability in the boundary layer and free troposphere. All effects of aerosols on Earth's climate are determined by a combination of their composition, size distribution and vertical distribution.

To better assess the role of mineral dust in the climate system, it is therefore needed to observe their composition and distribution, vertically as well as horizontally, and analyze its transport and emission processes. Ground-based measurement stations typically offer high quality observations of those aerosol parameters, but have poor horizontal resolution. Due to the high spatial and temporal variability of mineral dust events, remote sensing from space is the most adequate tool to daily monitor them at global scale.

A large effort has already been made to develop satellite products for retrieving aerosol properties. The total aerosol columnar load, expressed in aerosol optical depth (AOD) or optical thickness (AOT) is a parameter that many sensors provide, such as Moderate Resolution Imaging Spectroradiometer (MODIS, Remer et al. (2005); Levy et al. (2013)), Advanced Along-Track Scanning Radiometer (AATSR, Veefkind et al. (1998)), POLarization and Directionality of the Earth's Reflectances (POLDER, Deuzé et al. (2001)), Ozone monitoring instrument (OMI, Torres et al. (2013)) and Visible Infrared Imaging Radiometer Suite (VIIRS, Jackson et al. (2013)). They measure in the UV, visible or near-infrared and typically report the AOD around 550 nm. Generally, those instruments also offer additional information on aerosol size, type or optical properties. However, measurements made in the UV, visible or near-infrared are limited to daytime observations and often have difficulties retrieving aerosol properties over bright surfaces such as deserts (Xu et al., 2018). Moreover they don't provide information on the effect of mineral dust on the longwave thermal radiation, crucial for understanding the total aerosol radiative forcing.

Hence recently, also infrared sensors are used to retrieve aerosol properties. Further, these sensors allow making observations

at nighttime. Currently, global long term data sets of AOD are available from infrared sensors like the Atmospheric InfraRed Sounder (AIRS) and the Infrared Atmospheric Sounding Interferometer (IASI), onboard the polar-orbiting Aqua and MetOp satellites, respectively (DeSouza-Machado et al., 2010; Capelle et al., 2018; Clarisse et al., 2019; Popp et al., 2016). They can additionally provide dust layer mean altitude because infrared channels are sensitive to different levels of the atmosphere. Van-
5 denbussche et al. (2013) have developed a strategy to retrieve aerosol profiles at seven distinct heights using thermal infrared (TIR) radiances from the hyperspectral IASI sensor. Thereby providing additional information on the daily 3-D dust distribution on global scale. This retrieval algorithm is called MAPIR (Mineral Aerosol Profiling from Infrared Radiances, Popp et al. (2016)) and is based on Rodgers optimal estimation method (Rodgers, 2000). Also Cuesta et al. (2015) developed a method to derive dust extinction profiles with 1 km resolution at 10 μm from IASI. The main differences between MAPIR and this alter-
10 native study are that Cuesta et al. (2015) follow an auto-adaptive Tikhonov–Phillips–type approach and their method has until now only been applied to a very limited number of dust events, while MAPIR provides a global data set over a long time period using optimal estimation However, higher resolution aerosol profiles are only available with the use of active lidar instruments, such as the Cloud–Aerosol LIdar with Orthogonal Polarization (CALIOP) onboard CALIPSO. This two-wavelength (532 nm and 1064 nm) polarization-sensitive lidar provides products of aerosol backscatter and extinction with a vertical resolution of
15 30 m below 8.2 km and a horizontal footprint of 70 m. Due to this small footprint, it takes 16 days to scan the whole globe once and therefore the spatial and temporal coverage of CALIOP is unfortunately much more limited than that of IASI, which offers almost global coverage twice a day. Thus, with CALIOP it is highly likely that many mineral dust events are missed and it is therefore important to keep investing in the improvement of passive remote sensing retrievals.

Previous versions of the MAPIR algorithm often failed to retrieve mineral dust over desert surfaces with low emissivity due
to non-convergence or quality issues. To cope with those weaknesses and to make the processing less costly, a new version of MAPIR has been developed: version 4.1. In this manuscript the updated algorithm is presented and evaluated, the work is organized as follows. First an introduction to the IASI instrument is given in Sect. 2, together with a description of the instruments that were used to evaluate the retrieved profiles.. Section 3 contains the theoretical description of the retrieval method, the input parameters and the forward model used. Afterwards, in Sect. 4 the results of the processing of more than 10 years
IASI measurements are discussed together with an error analysis, followed by a comparison with measurements from other instruments in Sect. 5 to provide a quality assessment.

## 2 Instruments

This study uses the data of various instruments. The spectra on which the retrievals are carried out are from IASI. The evaluation of the retrieved profiles in Sect. 5 uses products derived from AERONET, CALIOP, CATS and two ground-based lidar
instruments. In this section we describe these instruments and their data selection.

## 2.1 IASI

The Infrared Atmospheric Sounding Interferometer (IASI) is a high-resolution TIR Fourier transform spectrometer onboard MetOp-A, -B and -C satellites, launched in October 2006, September 2012 and November 2018 respectively. It is set up to provide detailed observations of the global atmosphere for a period up to 15 years. Moreover, the IASI-NG instrument, which will have higher resolution and better signal-to-noise ratio, will be onboard of the MetOp-SG satellites which are to be launched between 2021 and 2035, guaranteeing continuous data up to 2040. They are on a polar sun-synchronous orbit about 817 km above Earth with equator crossing at 09:30 (21:30) mean local solar time in descending (ascending) mode, leading to an almost global coverage twice a day per instrument. IASI is a nadir viewing instrument with a swath width of 2200 km (off-nadir measurements with a viewing angle up to $48.3°$ on both sides of the satellite track) that scans in 30 elementary fields of view, each composed of 4 circular pixels of 12 km ground diameter at nadir and up to an ellipse of 39 km by 20 km at the extremities of the swath. It measures radiances over a spectral range that extends from $645~\text{cm}^{-1}$ to $2760~\text{cm}^{-1}$ with a spectral resolution of $0.5~\text{cm}^{-1}$ after apodization and has a radiometric noise of 0.2 K in the TIR atmospheric window (Clerbaux et al., 2009). Each spectrum is sampled every $0.25~\text{cm}^{-1}$, providing a total of 8461 radiance channels. In the TIR part of the IASI spectrum, as far as aerosols are concerned only mineral dust and volcanic ash have a significant spectral signature (e.g. Maes et al., 2016).

## 2.2 AERONET

AERONET (AErosol RObotic NETwork) is a worldwide network of around 400 permanently running ground-based sun photometers established by NASA and PHOTONS (LOA–CNRS) which measure atmospheric aerosol properties (Holben et al., 1998). The Cimel Electronique CE318 sun photometers perform measurements of sun irradiance in eight spectral bands (340, 380, 440, 500, 670, 870, 940 and 1020 nm ) every 15 minutes. For our comparisons in Sect. 5.1 we use the version 3 level 2.0 (cloud screened and quality-assured) Spectral Deconvolution Algorithm (SDA) retrieval of the coarse mode AOD at 500 nm. There is currently no aerosol type specification in the AERONET product, and the coarse mode mainly contains mineral dust, sea salt and/or volcanic ash.

## 2.3 CALIOP

The Cloud-Aerosol Lidar with Orthogonal Polarization (CALIOP) is an instrument on the Cloud-Aerosol Lidar and Infrared Pathfinder Satellite Observations (CALIPSO) platform (Winker et al., 2009), launched in 2006. CALIPSO is one of the six satellites in the A-train constellation, which are on a sun-synchronous polar orbit at about 705 km above Earth with equator crossing around 13:30 local time. CALIOP has two simultaneous co-aligned lasers at 532 and 1064 nm with a horizontal footprint of 70 m. It provides high resolution vertical profiles of aerosol attenuated backscatter and depolarization of which numerous data products can be derived, such as aerosol extinction profiles for six aerosol-types (clean marine, dust, polluted continental, clean continental, polluted dust and smoke). For the analysis in Sect. 5.2, the 5 km profile product from CALIOP data version V4-10 is used.

## 2.4 M'Bour lidar

The monoaxial Cimel Micro-Pulsed lidar is continuously operating at the M'Bour site (14.39° N, 16.96° W) close to Dakar, Senegal since 2005. This site is situated in a nature reserve, at less than 100 m from the Atlantic Ocean. The lidar provides attenuated backscatter profiles at 532 nm up to a height of 30 km, with a vertical resolution of 15 m. The extinction profiles are then calculated with 15 min averaged backscatter profiles co-located sun photometer, which is included in AERONET (AErosol RObotic NETwork). Hence, the lidar ratio can be retrieved and the related uncertainty reduced. More details on the instrument and the used inversion method can be found in Mortier et al. (2016).

Only cloud-free data are used for the analysis in Sect. 5.3.1 and in order to separate dust profiles from others we only use those profiles where the Angstrom exponent is lower than 0.4. Indeed, in Johnson and Osborne (2011) it is shown that the Angstrom exponent is typically lower than 0.2 for dust during the GERBILS campaign over the western region of the Sahara, but with measured values up to 0.6. Our selected threshold of 0.4 is a good compromise to be conservative enough but avoid the fine mode biomass burning and smoke aerosols which are plausible in M'Bour during the winter.

## 2.5 Al Dhaid lidar

The Multi-wavelength Raman polarization lidar PollyXT performed continuous measurements from March to April 2018 at the Al Dhaid site (25.24° N, 55.98° E) in United Arab Emirates. This is a rural site located at a deserted area, about 70 km east from Dubai, and 10 km from Al Dhaid town. To the east the site faces some hills/mountains (20 km away) and the sea (Gulf of Oman) at about 40 km distance.

PollyXT enables the retrieval of aerosol optical properties at three wavelengths, with an initial vertical resolution of 7.5 m along the line of sight and an initial temporal resolution of 30 s. More details of the instrument can be found in Althausen et al. (2009) and Engelmann et al. (2016).

To evaluate the MAPIR profiles in Sect. 5.3.2, the Klett inversion method (Klett, 1981) is applied to retrieve the aerosol backscatter coefficients and aerosol extinction coefficients at 355 nm and 532 nm, using lidar ratios of $\sim 45$ sr and $\sim 35$ sr, which are derived using Raman inversion for night-time lidar measurements (method description in Ansmann et al. (1990); Shang et al. (2018)). The volumetric depolarization ratio (VDR) and linear particle depolarization ratio (PDR) at 355 nm and 532 nm are also derived following the procedure described in Chazette et al. (2012).

We separate the optical properties of desert dust and the non-dust particles as a function of height, by applying the methodology proposed by Tesche et al. (2009). According to the literature (e.g. Groß et al. (2015); Tesche et al. (2009) and references therein), the PDR of dust is assumed to be 0.30 at 355 nm and 0.35 at 532 nm, with the non-dust PDR of 2 % and 3 %, respectively. Due to the lidar overlap effect the lower range limit is at $\sim 180$ m (bin 24) (Engelmann et al., 2016), the values below are filled with the average of the 23rd to 25th bin. The final dust optical properties used in the comparisons were vertically smoothed by the sliding averaging of 11 bins ($\sim 82$ m) and temporally averaged by 1 hour.

## 2.6 CATS

CATS is a lidar instrument onboard the International Space Station (ISS) providing vertically resolved cloud and aerosol properties at 1064 nm from March 2015 until October 2017 (Yorks et al., 2016). CATS is orbiting between 375 km and 435 km above Earth's surface at a 51.6° inclination with nearly a three-day repeat cycle (McGill et al., 2015). Due to this unique orbit path, the same location is not measured at the same local time every day by CATS, unlike sun-synchronous orbiting satellites like MetOp or CALIPSO.

For the study in Sect. 5.3.3, the Level 2 Operational extinction profiles between 0 and 5 km from version 3.00 are used, thereby selecting only the dust aerosol types (Dust, Dust Mixture and Marine Mixture). Quality filtering of the CATS data is done similar as in Lee et al. (2018).

## 3 Retrieval algorithm

This section presents the technical details of MAPIR v4.1, which are implemented in Python. MAPIR retrieves vertical profiles of desert dust concentration. It is an application of Rodgers' optimal estimation method (OEM) which is briefly described in the first subsection. Afterwards, the choice and set-up of the forward model is described, followed by a summary of how the state vector and observation vector are composed, together with their prior constraints.

### 3.1 Method

We use the notation and concepts of the optimal estimation approach as described by Rodgers (2000). The IASI observations are represented by an $m$-dimensional vector $\boldsymbol{y}$ and the unknown atmospheric state by an $n$-dimensional vector $\boldsymbol{x}$. The details of vectors $\boldsymbol{x}$ and $\boldsymbol{y}$ will be discussed in subsections 3.3 and 3.5, respectively. The relationship between $\boldsymbol{x}$ and $\boldsymbol{y}$ can be expressed as:

$$\boldsymbol{y} = \boldsymbol{F}(\boldsymbol{x}, \boldsymbol{b}) + \boldsymbol{\epsilon}, \tag{1}$$

where $\boldsymbol{F}$ is the forward model, $\boldsymbol{b}$ a set of fixed model parameters and $\boldsymbol{\epsilon}$ an error vector representing both model and measurement errors. When a description of the atmospheric state is given, the forward model computes the radiances at the top of the atmosphere as it would be measured by the IASI instrument. The radiative transfer model used here is Radiative Transfer for TOVS (RTTOV), which will be described in more detail in subsection 3.2. The inverse problem consists of finding a state vector that matches the observation well enough. By comparing the simulated spectra with the observed, a solution $\hat{\boldsymbol{x}}$ for the inverse problem can be found. Since the inversion problem is ill determined, additional constraints on the prior information are necessary and $\hat{\boldsymbol{x}}$ is found by minimizing a cost function $\chi^2$ determined by:

$$\chi^2 = [\boldsymbol{y} - \boldsymbol{F}(\boldsymbol{x}, \boldsymbol{b})]^T \boldsymbol{S}_\epsilon^{-1} [\boldsymbol{y} - \boldsymbol{F}(\boldsymbol{x}, \boldsymbol{b})] + [\boldsymbol{x} - \boldsymbol{x_a}]^T \boldsymbol{S}_a^{-1} [\boldsymbol{x} - \boldsymbol{x_a}]. \tag{2}$$

In the above expression, $\boldsymbol{x_a}$ is the a priori state vector, $\boldsymbol{S_a}$ the corresponding $n{\times}n$ covariance matrix and $\boldsymbol{S_\epsilon}$ the $m{\times}m$ measurement covariance matrix. As the forward model $\boldsymbol{F}(\boldsymbol{x}, \boldsymbol{b})$ is a complicated and non-linear function of $\boldsymbol{x}$, an iteration

method is required to obtain the minimum of this cost function. To ensure reaching closer to the minimum in each iteration step, the Levenberg–Marquardt modification of the Gauss–Newton method is adopted (Rodgers, 2000; Levenberg, 1944; Marquardt, 1963). This is a new aspect with respect to previous MAPIR versions, where the ordinary Gauss–Newton iteration method was used. Each step can then be described as follows:

$$5 \quad x_{i+1} = x_i + \left((1+\gamma)S_a^{-1} + K_i^T S_\epsilon^{-1} K_i\right)^{-1} \left(K_i^T S_\epsilon^{-1}(y - F(x_i)) - S_a^{-1}(x_i - x_a)\right), \tag{3}$$

where $\gamma$ is a damping parameter that changes every iteration step and $K$ is the weighting function matrix, or Jacobian, $K = \frac{\partial y}{\partial x}$. The parameter $\gamma$ starts at a value of 1 and is adapted in every step: if the cost function of the new state vector $x_{i+1}$ has increased relative to the cost function at the previous step ($\chi^2(x_{i+1}) > \chi^2(x_i)$), then the iteration step is repeated with $\gamma' = 10\gamma$. In case the new cost function has decreased, the new state vector will be accepted and $\gamma$ will be reduced with a factor of 2.

The iterations are stopped when the steps both in state space and measurement space are small enough or after 20 steps, whereby the retrieval is signalled as unsuccessful. The convergence criteria on the step sizes is taken from Rodgers (2000, p90), with $\epsilon = 10^{-1}$.

## 3.2 The forward model

The optimal estimation method requires a forward model that defines the relation between the state vector $x$ and the obser-
15 vation $y$. The radiances as measured by the IASI instrument can be simulated by the fast radiative transfer model RTTOV v12.1 (Radiative Transfer for TOVS), developed by the EUMETSAT Satellite Application Facility on Numerical Weather Prediction (NWP SAF). It consists of a predictor-based regression scheme, generated from a database of accurate line-by-line transmittances computed for a set of diverse atmospheric profiles (Saunders et al., 2017). The coefficients for the optical depths regressions are stored in instrument-specific coefficient files. We use the IASI v9 predictor coefficients calculated on 101 levels.
The aerosols effect is calculated with the Discrete Ordinate Method (DOM). As RTTOV is fast, easy to use, and allows for the computation of the Jacobians (the gradient of the radiances with respect to the state vector), it is very suitable for our retrieval approach. Especially, it is much faster than LIDORT (Spurr, 2008), which was used in previous MAPIR versions. The inputs to the radiative transfer model are presented below.

To compute the top of the atmosphere radiances in each of the IASI channels, atmospheric profiles of temperature, water vapour and aerosols are needed together with surface parameters and a viewing geometry. The profiles of other atmospheric gases are taken from the suitable reference profiles that RTTOV provides.

The atmospheric profiles of temperature and water vapour are taken from IASI level 2 operational products from EUMETSAT. As no full reprocessing has been done yet, this data is available in different versions (4 to 6), with version 5 and 6 starting at
30 14 September 2010 and 30 September 2014, respectively. From version 6 a new retrieval method was used which additionally includes microwave information. It should be noted that we have observed large quality differences in our retrieved aerosol profiles between these versions, as will be further discussed in Sect. 4. Indeed, as the temperature profile is an essential parameter in infrared retrievals, it has a major impact on our results.

The aerosol a priori concentrations are discussed in section 3.4. In addition, the radiative transfer model requires some microphysical properties of the aerosols. We have chosen to maintain the parameters used in previous versions of MAPIR (Vandenbussche et al., 2013): a log-normal particle size distribution (PSD) with median radius $0.6\,\mu m$ and geometric standard deviation of 2, corresponding with an effective radius of $2\,\mu m$, and the spatially invariant and time-constant refractive index of the GEISA–HITRAN dust-like data set, gathered by Massie (1994); Massie and Goldman (2003) from measurements by Volz (1972, 1973) and Shettle and Fenn (1979) on transported Saharan dust.

MAPIR v4.1 is based on thermal infrared radiances, therefore the surface parameters - surface emissivity and surface temperature - are of considerable importance for the modelled spectrum, especially over desert areas. The surface temperature is included in the retrieval (Sect. 3.3), while the surface emissivity is taken from two different databases, one for ocean and one for land surface. The ocean is very close to being a black body whereby its surface emissivity is close to 1, with a slight spectral variation. In that case, we use the emissivity of Newman et al. (2005). However, over land, there is a bigger variability. The surface emissivity varies spectrally and slowly as a function of time, depending on the surface composition, humidity and vegetation. Therefore, the emissivity database of Zhou et al. (2011), updated in 2015, is chosen for land surfaces. It is a monthly climatology at $0.25°$ horizontal resolution, obtained from IASI spectra. To retrieve the surface emissivity, Zhou et al. (2011) assume that the clear sky spectra (no clouds and no aerosols) coincide with the higher radiances within a month. It is therefore highly probable that, for places and times where dust is almost always present, the obtained emissivity is biased to low values.

## 3.3 State vector

The state vector contains those input parameters of the forward model that will be optimized to fit the observation. As mentioned in the above section, the surface temperature is included in the state vector because this is a dominant parameter for TIR radiation. Together with the aerosol load relative to the a priori concentration in the lowest seven layers of the troposphere, more specifically at $0.5\,km$, $1.5\,km$, $2.5\,km$, $3.5\,km$, $4.5\,km$, $5.5\,km$ and $6.5\,km$, they form the state vector of parameters we want to retrieve from the IASI measurements. The aerosol abundances are represented by their mid-layer altitudes with respect to sea level. In cases where the surface elevation is higher than the mid-layer altitude, the corresponding abundances are put to zero.

To avoid nonphysical negative concentrations during the iteration process, which can not be handled correctly by RTTOV, MAPIR v4.1 uses the logarithm of the relative aerosol load in each layer in the iterations, which is transformed to absolute aerosol concentrations ($particles \cdot cm^{-3}$) after convergence. This was not done in previous versions of MAPIR, where they were manually put to zero. It induces different underlying constraints as now log-normal statistics are assumed instead of normal. Consequently, retrievals starting at low a priori concentrations will be more constrained then when starting at higher concentrations (Deeter et al., 2007). This also means that the calculation of the information content parameters will be impacted by the dust load, as will be seen in Sect. 4.2.

### 3.4 A priori

To retrieve a unique solution, the OEM requires a priori information of the state vector. This information is a crucial constraint to make the inverse problem soluble.

For the dust aerosol retrievals, a monthly climatology derived from CALIOP measurements between 2007 and 2014 by the National Observatory of Athens (Amiridis et al., 2013, 2015), is used. This 3-D database provides high-resolution dust extinction profiles at 532 nm globally on a $1° \times 1°$ horizontal grid. The extinction at 532 nm is then converted to concentration ($\text{particles} \cdot \text{cm}^{-3}$) using an extinction cross-section computed with a Mie code and the PSD and refractive index described above. To assure data in each grid cell and continuity between adjacent cells, a running mean of the data set is calculated along $5° \times 5°$.

The additional constraints on the inverse problem require an a priori covariance matrix $\boldsymbol{S_a}$. The diagonal elements are represented by the square of the standard deviation of the individual elements of the state vector $\boldsymbol{x}$. Those standard deviations are taken to be 50 % of the a priori concentrations and the off-diagonal elements are filled according to a vertical Gaussian correlation of 1 km length.

The a priori surface temperature (*Ts*) is taken from IASI level 2 data, or the ECMWF ERA interim reanalysis skin temperature for dates before 14 September 2010 as the IASI temperatures are too unrealistic before, in level 2 version 4. Due to the difference in heat capacity of land and ocean, the surface temperature over land varies much more over time. This effect is even greater in arid regions where the temperatures fluctuate highly during the day. Therefore we believe existing databases of ocean *Ts* are more reliable than land *Ts* and the standard deviation of *Ts* is set at 15 K over land and at 5 K over ocean surfaces.

### 3.5 Observation vector

The observation vector $\boldsymbol{y}$ contains the radiances as observed by IASI, in brightness temperature. To save computation time, $\boldsymbol{y}$ does not hold the complete spectrum, but only the radiances in three spectral bands: 905–927 $\text{cm}^{-1}$, 1098–1123 $\text{cm}^{-1}$ and 1202–1204 $\text{cm}^{-1}$. The selection of these wave numbers is based upon the sensitivity to retrieve mineral dust profiles and is discussed in Vandenbussche et al. (2013). Together with the observation vector, a measurement covariance matrix $\boldsymbol{S_\epsilon}$ is defined. Although the reported spectral noise is 0.2 K (Clerbaux et al., 2009), we increase this instrumental error by a factor of 5, thus use $\boldsymbol{S_\epsilon} = \boldsymbol{I}$, to also take into account the uncertainties of the forward model and input parameters which are currently not modelled.

Only cloud-free observations can be used for the retrieval. To filter out the cloud spectra, the IASI operational level 2 cloud product is used with a threshold limit of 10 %. We have observed that dense aerosol scenes are occasionally misflagged as clouds within this product, for example the center of a big dust plume. It is important to note that this will lead to some discarded IASI scenes where actually there was a huge amount of dust.

## 4 Results

In the context of the C3S aerosols project, more specifically Copernicus Climate Change Service C3S_312a Lot 5, more than 10 years of IASI data have been processed. This data set starts at 25 September 2007, ends at 30 June 2018 and allows us to assess the quality of MAPIR v4.1. The processing is continued within C3S_312b Lot 2, every six months. For example, the retrievals until December 2018 will be delivered in February 2019. For technical reasons linked to the unavailability of the IASI spectra under the principal components scores format before 22 February 2011, only a part of the globe has been processed for that period: the so-called dust belt with longitudes between 80° W and 120° E and latitudes between 0° N and 40° N. From that date on, the IASI spectra are available in principal component scores and the whole globe is processed. However the latitudes above 60° N and below 60° S, where generally no desert dust aerosols are present, were neglected to save computational resources. Currently, only the data from IASI on MetOp-A has been processed.

To additionally reduce the computational power needed for this large data set, we applied a dust filter before undertaking the retrievals. To avoid missing too many dust events, we ran all retrievals in the dust belt area given in Fig. 1 and defined as follows: latitude between 5° S and 45° N, longitude between 20° W and 120° E and latitude between 5° S and 30° N, longitude between 80° W and 20° W. Outside this area we always performed the retrieval when the surface emissivity was below 0.85 in

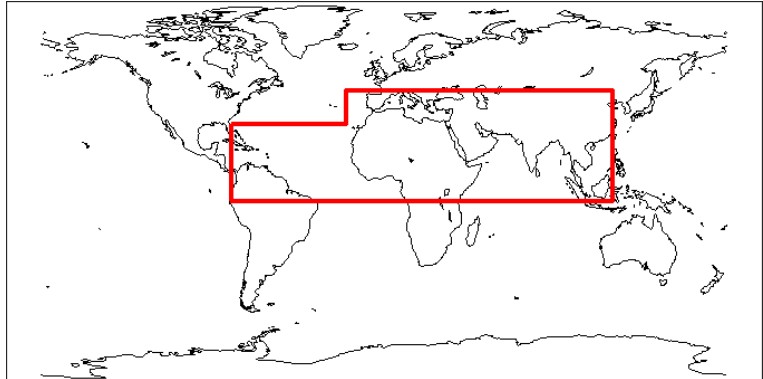

**Figure 1.** Map of the world with the red box defining the dust belt area where all MAPIR retrievals are undertaken (if cloud fraction is smaller than 10 % and no problem with input level 1 and level 2 data is detected).

any channel because those are potential desert areas. When the surface emissivity was higher than 0.85 the MAPIR retrievals were only performed when the following criterion on the slope of the spectrum was met:

$$BT_{1155-1160} - BT_{1082-1087} > 0.5 \text{ K}, \tag{4}$$

where BT stands for Brightness Temperature, $BT_{1155-1160}$ is the average BT between 1155 and 1160 $\text{cm}^{-1}$, and similarly for the second wave numbers range.

This large set of almost 11 years of MAPIR data allows us to perform reliable statistics to determine the value of the updated retrieval algorithm MAPIR v4.1. In the following the general performance of the retrieval will be discussed, followed by an

analysis of the information content and an example of MAPIR v4.1 output.

## 4.1 General performance of the retrieval

The quality of the retrievals can be described using different parameters. Here we will evaluate the quality filter, convergence
rate and the spectral residuals to get an idea of the overall performance of MAPIR v4.1.

To discard unreliable results, we apply a post-retrieval quality check with the following criteria: the root mean square of the
spectral residuals (RMSSR), being the difference between the modeled spectrum with the final state vector and the observed
spectrum, must be smaller than 1 K (which is about 5 times the IASI radiometric noise in the TIR); the 10 μm AOD must
be below 5 and the retrieved surface temperature ($Ts$) should be between 200 K and 350 K. The criteria on AOD and $Ts$ are
mainly to avoid cloudy scenes which were not detected by the IASI operational level 2 cloud product. We find that 84 % of
the retrievals pass the post-retrieval quality check and thus are said to be of good quality. There is good coverage of the Sahara
and Sahel regions, while this was one of the main shortcomings in earlier MAPIR versions. Indeed, when we apply the same
quality filter on the data set produced with MAPIR v3 under the European Space Agency aerosols Climate Change Initiative
phase 2 (Popp et al., 2016), only 59 % is accepted. Even though this previous data set covers only the dust belt region up to
December 2016, it is clear that MAPIR v4.1 performs better. Further, we observe an increase of quality through the time series,
with a yearly ratio of good quality retrievals between 71 % and 85 % until 2014 and above 90 % after 2014. This is probably
due to the different versions (and quality) of the EUMETSAT IASI l2 products for temperature profiles that are used for the
retrievals (see in Sect 3.2) of which the most recent, version 6, starts at 30 September 2014. Indeed, the temperature profile is
crucial for computing the radiative impact of dust aerosols, and biased temperature profiles certainly lead to biased dust aerosol
profiles.

As previously mentioned, we want the iteration scheme to find convergence within 20 steps. If this is not the case, the retrieval
is killed and flagged as failed. We do this mainly for computational reasons, but also because in those cases it is very likely
that the assumed ancillary data, such as surface emissivity, temperature or aerosol properties, are too far from reality. The con-
vergence rate was improved in MAPIR v4.1 by including the Levenberg–Marquardt modification. We see that only 0.6 % of
all attempted retrievals has to be stopped after 20 iterations, which is less than the 0.78 % with MAPIR v3. Those that fail are
likely cloudy scenes that were not correctly filtered out, or scenes in which the real situation was not well enough represented
by the used parameters. The retrievals were usually completed after two iteration steps. For the good quality retrievals, we
observe an average amount of iterations of 2.93 and a median of 2. Only 5 % of the retrievals that pass the quality filter needed
more than 6 iteration steps to converge.

To assess the quality of the converging retrievals that pass our aforementioned quality filter, we look at the values of the
RMSSR. Due to the quality filter, they are all between 0 K and 1 K, but there are more residuals close to 0 K as the median
RMSSR is 0.32 K. Furthermore we see a mean of 0.39 ($\sigma^2 = 0.05$). Overall, the observed spectra are well reproduced by the
simulated spectra, within error bounds.

## 4.2 Information content

To correctly interpret and use this data set of dust profiles, it is necessary to also consider the averaging kernels and degrees of freedom. The averaging kernels (AK) represent the vertical sensitivity of the retrieved profiles while the degrees of freedom (DOF), which is the trace of the AK matrix, give an estimate of the number of independent pieces of information that is contained in the measurement.

Rodgers OEM provides a way to calculate the AK matrices (Rodgers, 2000), but as we implemented the Levenberg–Marquardt (LM) method, this computation has to be adapted. Ceccherini and Ridolfi (2010) give a detailed description of how to deal with the AK matrix in such cases. It takes into account both the LM damping term $\gamma$ and all the iteration steps that were required to reach the minimum of the cost function. They are calculated as follows:

$$A = T_r K, \tag{5}$$

where $K$ is the Jacobian matrix of the forward model with respect to the state vector in the true profile and $T_r$ is a recursively calculated matrix which depends on the path in the parameter space followed by the minimization procedure. The recursive formula for the matrices $T_i$ is given by:

$$\begin{cases} T_0 & = 0 \\ T_{i+1} & = S_i K_i^T S_\epsilon^{-1} + (I - S_i K_i^T S_\epsilon^{-1} K_i - S_i S_a^{-1}) T_i \end{cases} \tag{6}$$

with $S_i = S_a^{-1} + K_i^T S_\epsilon^{-1} K_i$ and $K_i$ the Jacobian with respect to the state vector at step $i$.

The shape of the averaging kernels are quite variable, an example is given in Fig. 2. Two profiles are given together with their a priori, averaging kernels and degrees of freedom. The shape of the AKs includes information on the vertical resolution of the retrieval. In Fig. 2 we see they are quite broad, with overlapping peak altitudes, which suggests that adjacent retrieved aerosol concentrations are correlated. Besides, the peaks of the AK clearly coincide with the retrieved aerosol peaks. Indeed, as noted by Deeter et al. (2007), the averaging kernels tend to be smaller where there are low concentrations, and larger at high aerosol concentrations, as a consequence of using a logarithmic state vector. When summing up the diagonal elements of $A$, we get a value for the degrees of freedom for signal. It describes the number of independent pieces of information that can be retrieved from the observation. However, due to the different underlying constraints when performing log-normal retrievals, retrievals in very dusty regions are less constrained by the a priori and relatively more sensitive to the true profile, thereby increasing the DOFs (Deeter et al., 2007). Indeed, in the more dusty scenes we observe a median DOF of $1.4$ and a mean of $1.43$ ($\sigma^2 = 0.15$). In clear regions, the DOFs can be very low, as also illustrated in the next section .

## 4.3 Global distribution

MAPIR v4.1 results for both the morning and evening overpass on 9 June 2018 are presented in Fig. 3. Maps of the AOD at 10 µm are plotted together with the corresponding DOF and RMSSR. Areas on the AOD maps with missing data correspond

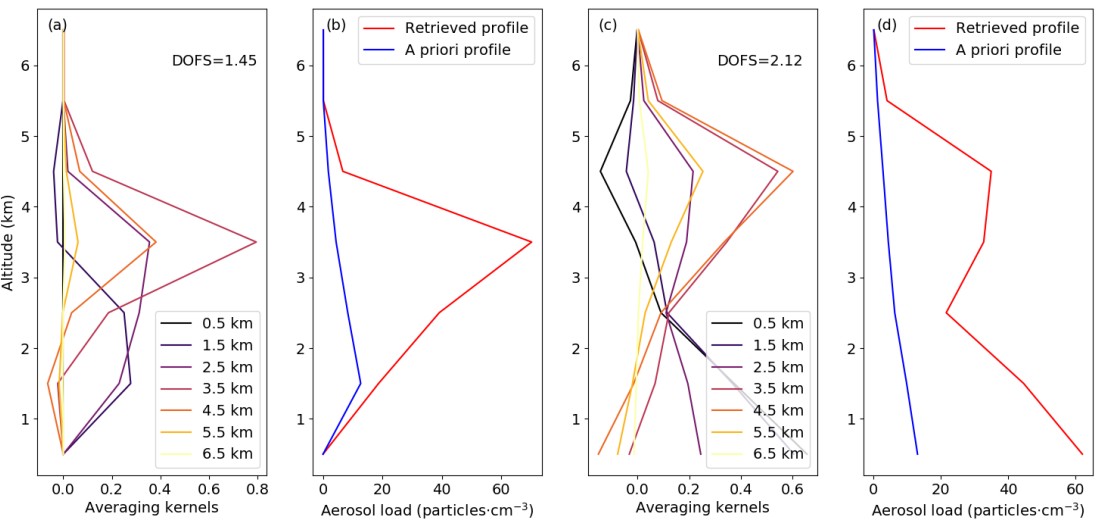

**Figure 2.** Averaging kernels and associated retrieved profile at two different locations: (a,b) 14.97° N, 23.7° E on 12 March 2016 and (c,d) 28.99° N, 5.3° E on 27 May 2016. Degrees of freedom (DOFs) are also given.

to areas which were identified as cloudy by the IASI operational level 2 cloud product, areas where the retrieval didn't pass our quality filter or where there was no IASI data. On the DOF and RMSSR maps also areas that were not treated due to our complex dust filter discussed in the introduction of this section are omitted. In the AOD map, those areas are considered to have an AOD of 0.

To calculate the MAPIR 10 μm AOD, we sum the aerosol concentrations ($particles \cdot cm^{-3}$) in each layer multiplied by its thickness and multiply this with the extinction cross-section at 10 μm as calculated with Mie theory. In Fig. 3(a) and 3(b) we can observe several dust events occurring on that day. Major amounts of dust are emitted in the center of the Sahara, while a plume is being transported from the Sahara desert over the Atlantic Ocean. The apparent discontinuity along the West coast of Africa (which can be better seen in Fig. 4), is probably caused by the shape of the dust plume itself. This event observed

by MODIS shows similar patterns (the data can be visualized at https://worldview.earthdata.nasa.gov). However, a small area near the South coast of Mauritania where the MAPIR AOD is low compared to its surroundings, shows relatively high RMSSR values almost reaching our quality filter of 1 K. This could indicate that MAPIR is slightly underestimating the dust load in that particular area. Near the coast of Oman, in the northern part of the Arabic Sea, another transported plume is visible on the AOD plots. Additionally, we also observe dust emissions in northern India and the Taklamakan desert during daytime and

around the southern part of the Red Sea during nighttime.

In 3(c) and 3(d) the DOFs are plotted for each retrieved aerosol profile. They are clearly connected to the AOD values: in areas with a high dust load the degrees of freedom go up to 2, while in clear areas we observe values close to 0. Indeed, as mentioned

before, the averaging kernels and DOFs are linked with the retrieved aerosol loads because of the constraints associated with using a log-normal state vector.

In 3(e) and 3(f) the global distribution of the RMSSR of the good quality retrievals on 9 June 2018 is presented. These values seem to be randomly distributed over the globe, not related to the dust load. However, the RMSSR are clearly smaller over ocean than over land. This is most likely due to the lower uncertainty on ocean surface properties compared to land.

Figure 4 shows a cross section of the retrieved dust distribution on the morning of 9 June 2018 to have more detail on the events

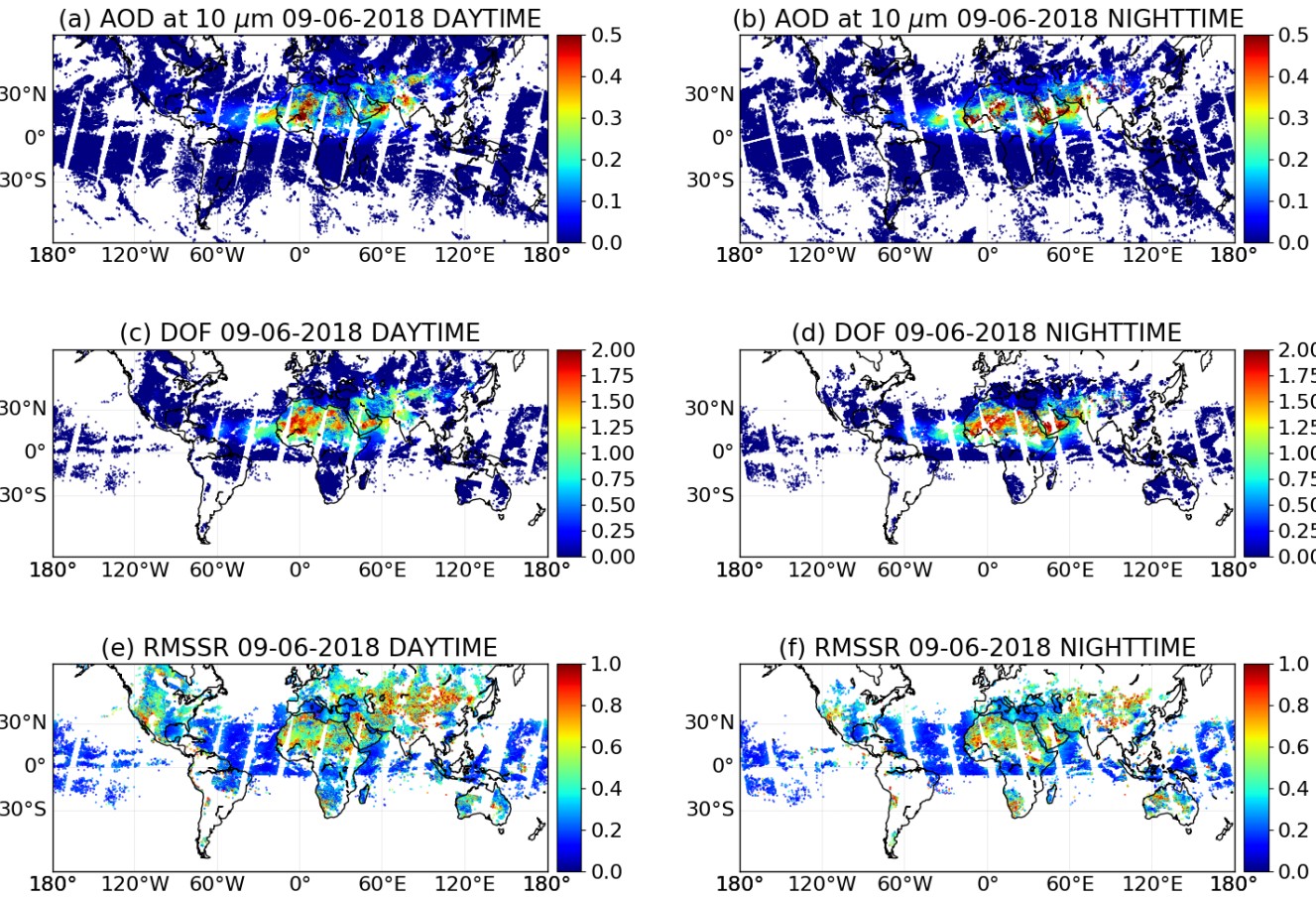

**Figure 3.** (a, b) Maps of the aerosol optical depth (AOD) at 10 μm on 9 June 2018, calculated by integrating the retrieved aerosol profiles and multiplying by the extinction cross-section. (c,d) Maps of the degrees of freedom (DOF), June 2018. (e,f) Maps of the root mean square of the spectral residuals (RMSSR). Daytime (nighttime) measurements on the left (right) correspond to a mean local solar time of 09:30 (21:30) when crossing the equator.

detected in Fig. 3(a). The plume over the Atlantic Ocean is indeed transported dust, at an elevated altitude of approximately 3–4 km. Over the Sahara desert, big amounts of mineral aerosols are emitted in the troposphere up to 5 km, with the highest

load near the surface. This suggests those areas are possible dust sources. Finally, the plume near the Gulf of Oman is spread over different altitudes. It is likely that the dust was emitted over land near the coastline (around 60° E) and then transported both over the Arabic Sea and landward over the Arabian peninsula.

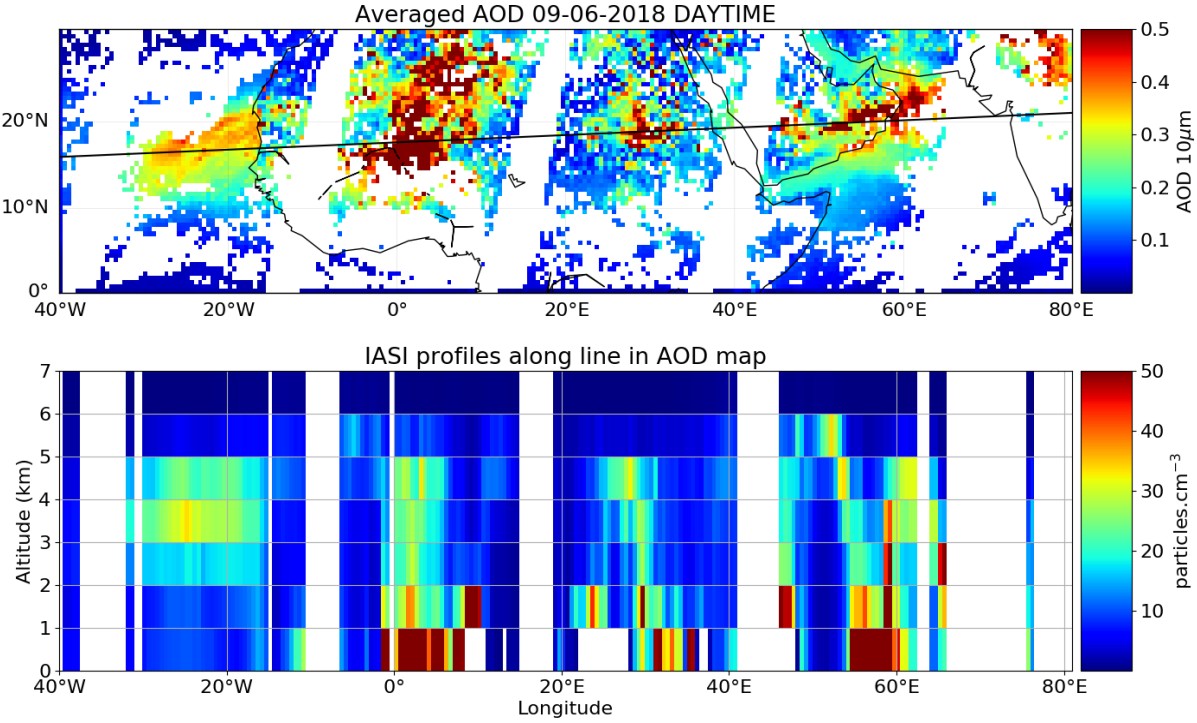

**Figure 4.** Cross section of MAPIR dust profiles on 9 June 2018. The first plot gives a map of the retrieved daytime 10 μm AOD on a 0.5° by 0.5° grid over the region we are interested in. The black line represents the locations along which the cross section in the second plot is given. Each profile of the cross section is an average of all profiles within 0.5° of the transect line.

## 5 Evaluation

5   To evaluate the MAPIR v4.1 dust profiles, comparisons with recognized independent data sets are needed. First we examine the AERONET data set, comparing the integrated profiles resulting in the AOD, as this is the most common reported dust feature. Second, we compare the mean altitude of the aerosol layer from MAPIR (i.e. altitude for which half the aerosols are below and half above) to the mean altitude of the dust aerosols from measurements by CALIOP onboard CALIPSO. As there is a time lag of 3 to 5 hours between IASI and CALIOP overpass times, a transport model is used to model the air mass movement
10   during that time. However, over dust source areas, this might not be sufficient, as an emission event could occur at the IASI overpass time and be finished at the CALIOP overpass time, with a part of the dust quickly deposited and a part transported. In that case, the two instruments would observe completely different air masses and vertical profiles of dust, and the transport

model may not be sufficient to account for that difference. Therefore, as final exercise, we provide a qualitative comparison of MAPIR dust profiles with other lidar measurements, close to dust source areas, for which a shorter time difference is possible (1 hour). These lidar measurements are from two ground-based instruments at M'Bour (Senegal) and Al Dhaid (United Arab Emirates), and from the Cloud–Aerosol Transport System (CATS) instrument onboard the International Space Station (ISS).

## 5.1 AOD evaluation with AERONET

We compare the MAPIR 10 μm AOD with in-situ measurements at AERONET sites. As AERONET provides only daytime measurements of AOD in the visible range of the spectrum, we have to be careful when comparing them to our thermal infrared AOD values. Therefore only the IASI measurements at local morning are used here and the MAPIR AOD values at 10 μm are converted to a visible equivalent at 550 nm using the ratio of extinction cross sections at both wavelengths. The values of these extinction cross sections are calculated according to Mie theory with the aerosol characteristics as mentioned in previous sections and are thus dependent on the chosen micro-physical properties. As we assume spherical particles with a fixed size distribution and refractive index, these micro-physical properties can deviate from reality and significant uncertainty is introduced by this conversion. Indeed, a sensitivity analysis performed by Capelle et al. (2014) shows the impact of the dust aerosol micro-physical properties on the infrared to visible conversion.

Comparisons are then made for those AERONET stations for which there is enough data to match the IASI measurements. There should be at least 100 matches over the whole period (from 25 September 2007 until 30 June 2018). The matches should be close both in time and space and are found as follows: we take those IASI measurements which are within 0.25° of an AERONET station, for each of them we take the AERONET measurement closest in time with maximum one hour time difference. Furthermore, and as also done by Capelle et al. (2018), we eliminate those matches for which the measured differences are beyond the 97th percentile as we believe these are caused by bad input data. By removing these questionable data we can better assess the true quality of the retrieval.

Finally, we considered only sites for which the median of the AERONET coarse mode AOD at 500 nm over the considered time period is higher than 0.05. As mentioned in Sect. 2.2, the coarse mode AOD contains all coarse mode aerosols, i.e. mainly dust, sea salt and volcanic ash. The selection therefore does not ensure the presence of only dust at those selected AERONET sites. This leads to a set of 72 stations spread over different regions. A list of the sites including their coordinates can be found in Appendix A.

For each of the 72 stations, we calculated the Pearson correlation coefficient between the AERONET AOD and the MAPIR AOD. Figure 5 shows the stations on a map with their corresponding correlation in color, the exact values can be found in Appendix A. Overall we see a strong agreement between MAPIR retrieved AOD and AERONET measured AOD. More than 93 % of the matched AERONET stations have a moderate ([0.4, 0.59]), strong ([0.6, 0.79]) or very strong ([0.8, 1.0]) positive correlation with the MAPIR retrieved AOD. Moreover we see a mean Pearson correlation coefficient of 0.66 on all stations. The coastal stations where the presence of sea salt aerosols plausibly impacts the AERONET coarse mode AOD and its correlation with MAPIR dust AOD are indicated with an asterisk in Table A1.

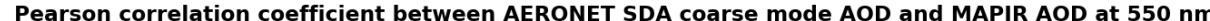

## Pearson correlation coefficient between AERONET SDA coarse mode AOD and MAPIR AOD at 550 nm

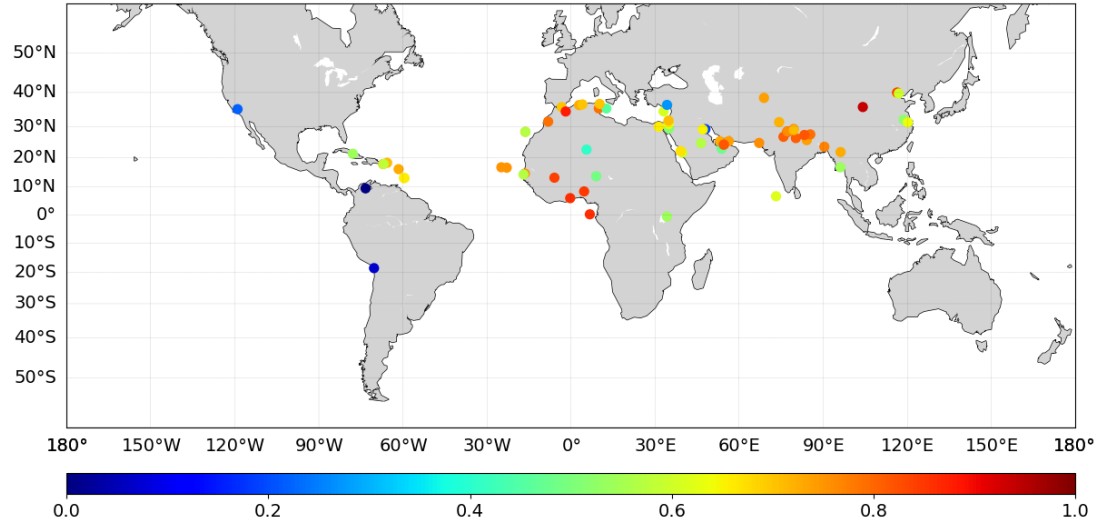

**Figure 5.** Map of the 72 AERONET stations that were matched with IASI measurements. The color scale represents the value of the Pearson correlation coefficient between the AERONET SDA coarse mode AOD and the MAPIR AOD converted to 550 nm.

In the northern part of India all stations have a strong or very strong Pearson correlation coefficient ranging from 0.71 at Pantnagar to 0.86 at Gual Pahari with a mean of 0.78 over 14 stations.

The region just North and South of the Sahara covers stations with an overall good correlation. There is for instance a very strong correlation of 0.86 at Sao Tome. However, two stations in the center of the Sahara and Sahel region have only a mod-
erate positive correlation with a coefficient of 0.41 and 0.49 at Tamanrasset and Zinder airport, respectively. The Tamanrasset AERONET station is located in the south foothills of the Hoggar Mountains in Algeria at almost 1400 m altitude, and at the northeast limit of a main source area (Schepanski et al., 2012; Ashpole and Washington, 2013; Todd and Cavazos-Guerra, 2016). It is therefore surrounded by very different air masses in different directions, which is expected to lead to noisy comparisons when using our simple criterion of distance between a IASI footprint center and the AERONET station. Zinder, on
the other hand, is a city in the south of Niger, and we see no reason on the location of the Zinder AERONET station to justify the lesser correlation. However, the station at Zinder seems to have almost only AERONET measurements during dust seasons and big events and very few from background situations, which can lead to biased statistics. Still it remains unclear why the retrieval shows such weak performance at higher aerosol loads near Zinder. As both Tamanrassat and Zinder AERONET stations are situated in the Sahara and Sahel deserts, this mismatch between AERONET AOD and MAPIR AOD could also point
to an incorrect surface emissivity there, as the used data set of Zhou et al. (2011) might be biased by the presence of aerosols in dusty regions.

The transport of the Saharan desert dust across the Atlantic Ocean is observed at several stations in the Caribbean, such as:

Camaguey, Guadeloup, Cape San Juan and La Parguera. With correlation coefficients ranging from $0.54$ to $0.73$, they show a moderate to strong correlation between AERONET and MAPIR AOD of coarse mode transported dust. However, those are coastal stations where the coarse mode probably contains sea salt aerosols with a possible impact on the AERONET coarse mode AOD.

Three of the sites with a weak or very weak correlation are situated in the American continent, in areas not known for the presence of dust: Arica, Bakersfield and UPC–GEAB–Valledupar. Arica is a coastal station, potentially experiencing sea salt aerosols. For the other 2 stations, the reason for the discrepancy is not clear.

Another AERONET station with a weak correlation is Kuwait University. As there is a second AERONET station very close by, Shagaya Park, which has a strong correlation of $0.66$, it is not immediately a sign of incorrect performance of the MAPIR

retrieval. Moreover, the AERONET data at Shagaya Park is from the period between 2015 and 2016 while the data at Kuwait University is from 2008 to 2010. This means this discrepancy could also be caused by the quality improvement of the water vapour and temperature profiles of the IASI operational l2 data over that time period. Indeed, the co-located retrievals at Shagaya Park use the better IASI l2v6 temperature profiles, while the retrievals at Kuwait university were computed using profiles from the IASI l2v4 product.

Apart from the Pearson correlation coefficient, we calculated a linear regression line for every station of which the slope and intersection of the Y-axis can be found in Appendix A. We see that overall the slope is around $1$, or slightly below, with a median of $0.71$. These lower values for the regression slope might indicate an underestimation of the conversion factor used for transforming the infrared AOD to its visible equivalent. The y-interception is almost everywhere close to $0$ with a median value of $0.06$.

To illustrate the similarities and differences between MAPIR AOD and AERONET AOD in an alternative way, Fig. 6 shows time series for the AOD at $4$ AERONET stations. The dates that are plotted are those for which AERONET data is available at that particular station. In agreement with the good correlation coefficients, we see that the MAPIR AOD reproduces the AERONET AOD well throughout the year at those sites. At sites like Masdar Institute and Koforida, the big AOD variation is reproduced by the MAPIR AOD. At Tunis Carthage, where there is in general a lower AOD, MAPIR sometimes misses a peak

in AOD leading to a small underestimation of the AOD by MAPIR at that station. Another way of showing this are the plots in the second column of Fig. 6. It presents a scatter plot of the AOD differences in function of the size of the AERONET AOD. Additionally, the data per station is split into AOD bins of equal quantity. Binned medians (black dots) and interquartile ranges (IQR, vertical black lines) of the AOD differences are shown on the plots. For example at Tunis Carthage, we see in the AOD difference scatter plot that most of the observations are low AOD cases with small positive bias, 4 out of the 5 AOD bins are

situated below $0.1$. The AOD bin of larger AOD values shows a slightly negative bias, thus they are generally underestimated. This negative trend, positive bias for low AOD and negative bias for higher AOD, is to some extent present at the other stations in Fig. 6 too.

Figure 7(a) shows the same kind of plot but for all AERONET stations combined and split up into $10$ bins of equal size. The total number of points used for these statistics are 76976, for the whole time period over the 72 selected AERONET stations.

The binned medians show that the low AOD cases (AOD < $0.1$) have a small positive bias, the cases with AERONET AOD

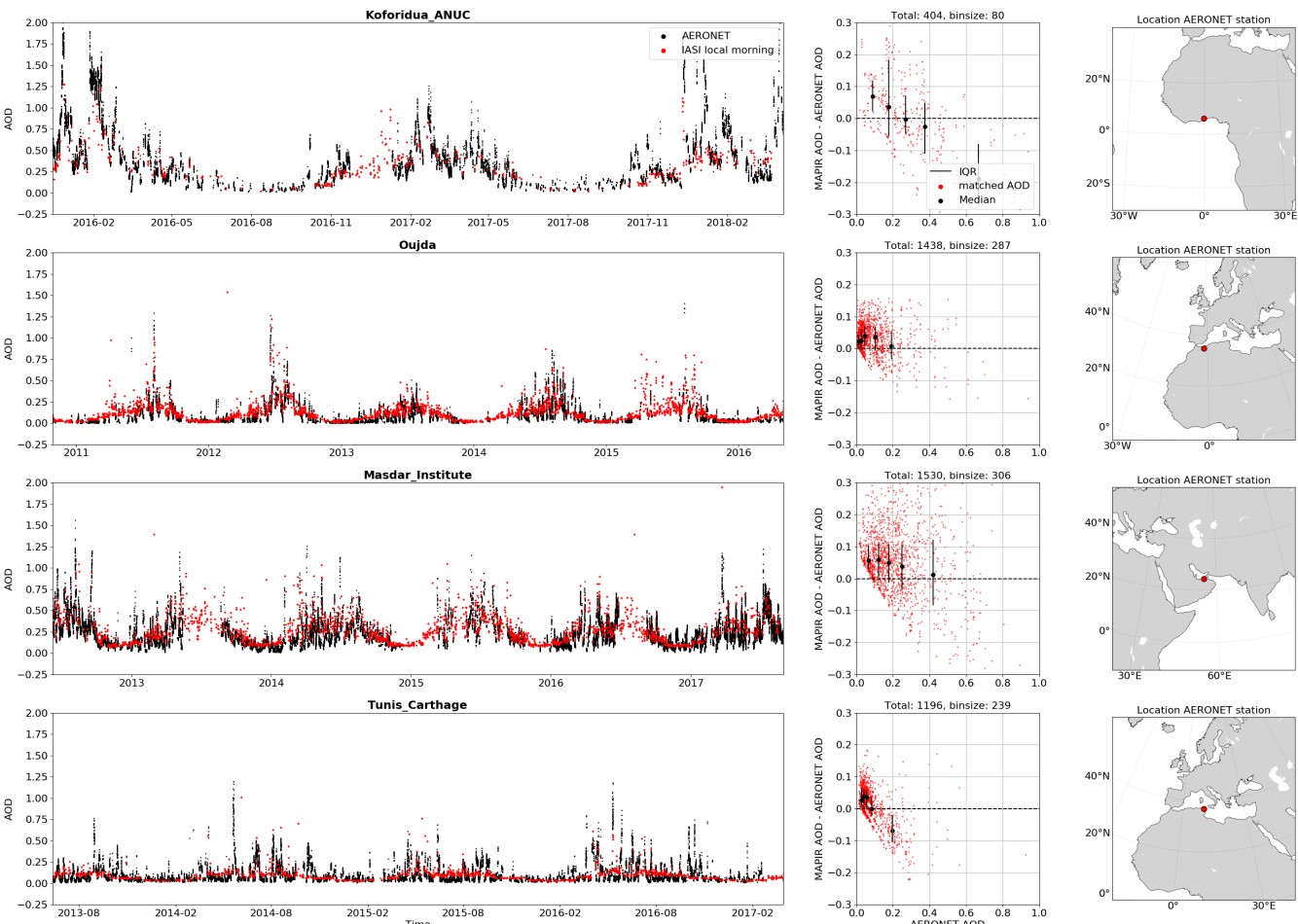

**Figure 6.** Time series of the available AERONET data at four stations in the first column: Koforidua ANUC, Oujda, Masdar Institute and Tunis Carthage. The black dots are the AERONET SDA coarse mode AOD, the red dots are the matched MAPIR AOD converted to 550 nm. The second column gives the corresponding scatter plots of the difference between the matched observations, together with the median and interquartile range (IQR) of 5 bins of equal size. The location of each AERONET site is given in the third column.

between 0.1 and 0.4 have almost no bias but there is a bigger spread, and the most dusty scenes (AOD > 0.4) show a small negative bias. The imaginary line that is visible in the lower left part of the scatter plot is due to the positivity constraint on the AOD values. This constraint is probably the main cause for the small positive bias for low AOD cases. Moreover, as the distribution of AOD is skewed to the right, it will affect the overall bias.

5    When calculating the mean difference of all matched AOD measurements, we observe a small positive bias of only 0.04 ($\sigma = 0.16$) with respect to AERONET. In Fig. 7(b) the associated difference histogram is given. The root mean square error (RMSE) between all matched AOD is 0.17 and more than 70% of the absolute differences fall below 0.1. These numbers show

that AOD values retrieved by MAPIR v4.1 are quite reliable and most importantly, MAPIR is improved with respect to its previous versions. Indeed, in Popp et al. (2016) a similar comparison with AERONET AOD was done using MAPIR v3. With a bias of 0.28, MAPIR v3 had a significant overestimation which is now almost gone.

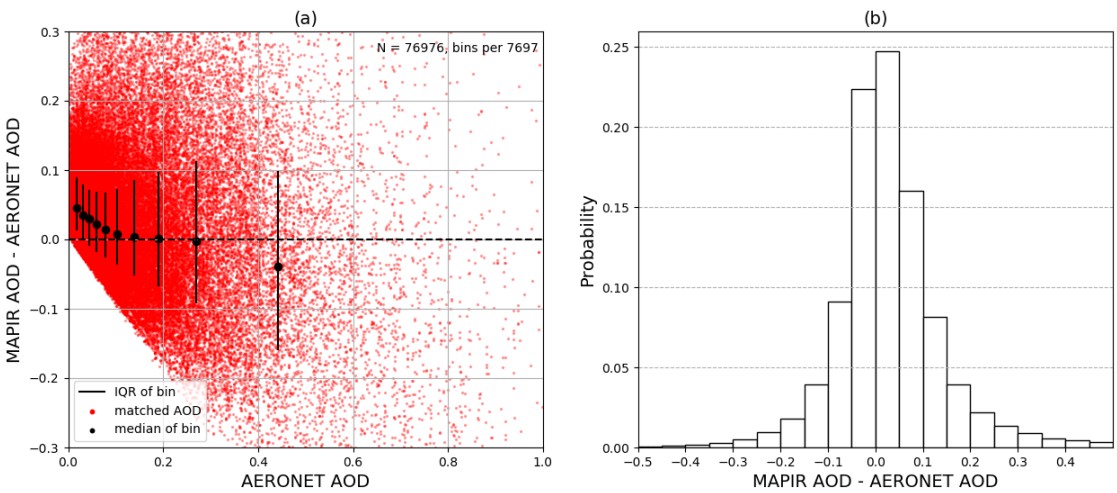

**Figure 7.** MAPIR AOD, converted to 550 nm, versus the AERONET SDA coarse mode AOD difference distribution. (a) Scatter plot of the AOD difference in function of the AERONET AOD (red dots), for all considered stations combined. Data are split into 10 bins of equal quantity, corresponding binned median (black dots) and interquartile range (IQR, vertical black lines) are shown. (b) AOD difference histogram.

## 5.2 Altitude evaluation with CALIOP

The aerosol altitude from the updated MAPIR algorithm was evaluated by comparing with altitudes from CALIOP. The fact that the MAPIR a priori is obtained from CALIOP measurements does not invalidate the MAPIR evaluation with CALIOP data. A monthly climatology over 8 years is used for the a priori (i.e. for each month, the mean profile from the same month from 8 years of CALIOP data), with a running mean over $5°$ latitude and longitude as detailed in section 3.4, while the evaluation is done by comparing single co-located measurements. The comparison was made following the methodology described in Kylling et al. (2018). Within the region of interest the closest CALIOP swaths in time and space to the MAPIR dust pixels were identified. Due to different equator crossing times between the CALIPSO and MetOp satellites and the possible transport of dust, a co-location criterion of maximum 5 h and 500 km was used for a first selection of CALIOP data. Only CALIOP data with vertically continuous dust profiles and cloud discrimination values between $-100$ and $-20$ (Winker et al., 2013) were retained for further analysis. CALIOP dust altitudes were calculated for the remaining profiles and moved in time and space to the IASI overpass time using the FLEXTRA trajectory model (Stohl et al., 1995). Finally, co-location of the CALIOP and MAPIR dust altitudes were checked and maximum differences of 20 km allowed. CALIOP profiles do not provide a unique

dust altitude. Here we use the same CALIOP altitudes as Kylling et al. (2018), namely the purely geometric mean altitude (mean of the bottom and top altitudes of the dust layer) and the cumulative extinction altitude (dust altitude set to altitude where the cumulative extinction at 532 nm is half of the total extinction column). The 5 km profile product from CALIOP data version V4-10 was used for the comparison. The comparison is made for two periods. The first period is identical to the same time and region used by Kylling et al. (2018): 18–27 March, 22 May–1 June, 1–12 July, and 14–20 September in 2010, totalling 40 days. These dates cover four desert dust events in the region between 0-40° N and 80° W-120° E. The second period covers four dust events in 2017: 21–30 April, plumes over Africa, Middle East and Asia, mainly over land; 3–12 July, large plume over Africa with massive transport to America, and some plumes over the Middle East and India; 1–10 October, some dust over Africa and some activity in the Taklamakan area, Middle East and India; and 21–30 December, Sahel plumes and East Asia dust. For 2017 the full region between 60° S–60° N and 180° W–180° E was included in the analysis. The findings for the MAPIR dust altitude comparison with CALIOP dust altitudes are summarized in Table 1. The table includes the 2010 data

**Table 1.** The mean ± the standard deviation of the dust altitude difference between the MAPIR and CALIOP dust altitudes, and the number (#) of co-located points. The inlay is the percentage of MAPIR altitudes that are within the CALIOP layer. MAPIR 3.2/3.4 results are taken from Kylling et al. (2018).

| Year | 2010 | | 2010 | | 2017 | |
|---|---|---|---|---|---|---|
| Algorithm | MAPIR v3.2/v3.4 | | MAPIR v4.1 | | MAPIR v4.1 | |
| CALIOP altitude | Cumulative extinction | Geometric mean | Cumulative extinction | Geometric mean | Cumulative extinction | Geometric mean |
| **CALIOP, All data (Day and Night, Ocean and Land)** | | | | | | |
| Altitude difference (km) | $0.590 \pm 1.213$ | $0.078 \pm 1.108$ | $-0.361 \pm 1.090$ | $-0.871 \pm 1.047$ | $-0.322 \pm 1.044$ | $-0.640 \pm 1.031$ |
| points (#) | 2620 | 2408 | 2575 | 2358 | 2304 | 2244 |
| inlay (%) | 83.1 | 81.1 | 79.8 | 77.9 | 77.2 | 76.6 |
| **CALIOP Day, Land** | | | | | | |
| Altitude difference (km) | $0.357 \pm 1.665$ | $0.087 \pm 1.572$ | $-0.567 \pm 1.535$ | $-0.888 \pm 1.435$ | $-0.452 \pm 1.160$ | $-0.633 \pm 1.146$ |
| points (#) | 605 | 598 | 607 | 597 | 1097 | 1100 |
| inlay (%) | 58.5 | 57.7 | 63.2 | 63.0 | 69.1 | 69.1 |
| **CALIOP Day, Ocean** | | | | | | |
| Altitude difference (km) | $0.783 \pm 0.913$ | $0.340 \pm 1.187$ | $-0.456 \pm 1.076$ | $-0.850 \pm 1.021$ | $-0.225 \pm 0.709$ | $-0.535 \pm 0.768$ |
| points (#) | 172 | 170 | 204 | 204 | 312 | 313 |
| inlay (%) | 74.4 | 72.4 | 58.5 | 57.6 | 72.8 | 71.9 |
| **CALIOP Night, Land** | | | | | | |
| Altitude difference (km) | $0.567 \pm 1.020$ | $0.038 \pm 0.903$ | $-0.314 \pm 0.920$ | $-0.822 \pm 0.896$ | $-0.181 \pm 1.064$ | $-0.625 \pm 1.017$ |
| points (#) | 1501 | 1330 | 1390 | 1228 | 689 | 661 |
| inlay (%) | 91.0 | 89.4 | 85.8 | 84.3 | 85.5 | 85.5 |
| **CALIOP Night, Ocean** | | | | | | |
| Altitude difference (km) | $1.008 \pm 0.741$ | $0.094 \pm 0.678$ | $-0.148 \pm 0.670$ | $-1.035 \pm 0.666$ | $-0.247 \pm 0.556$ | $-0.943 \pm 0.553$ |
| points (#) | 342 | 310 | 374 | 329 | 206 | 170 |
| inlay (%) | 96.5 | 95.8 | 96.2 | 93.9 | 99.5 | 100.0 |

from the comparison presented in Kylling et al. (2018) for MAPIR v3.2/v3.4.

For 2010 the previous MAPIR version in general overestimated both the cumulative extinction (by 0.357 to 1.008 km) and geometric mean (by 0.038 to 0.340 km) dust altitudes from CALIOP. MAPIR v4.1 generally underestimates the CALIOP dust

altitudes by $-0.148$ to $-0.567$ km (cumulative extinction) and by $-0.822$ to $-1.035$ km (geometric mean). The reason for this is most likely because the previous MAPIR version retrieved the dust concentrations on levels starting at an altitude of 1 km. MAPIR v4.1 retrieves layer concentrations where the lowest layer is between 0 and 1 km. Thus, the new MAPIR version will give a lower mean dust altitude compared to the previous versions. The difference between the cumulative extinction

and geometric mean altitude differences are about the same for both versions. However, MAPIR v4.1 gives a consistently smaller standard deviation by about $0.1$ km. The percentage of MAPIR altitudes within the CALIOP dust layer is somewhat smaller for MAPIR v4.1 and especially for CALIOP day time measurements over the ocean. However, for night time CALIOP measurements over the ocean MAPIR v4.1 places more dust altitude within the CALIOP dust layer.

For 2017 a smaller difference between MAPIR and CALIOP dust altitudes are observed compared to 2010. We also note that

the standard deviation for most cases (CALIOP night data over land being an exception) is smaller for the 2017 comparison. This may be due to the improved IASI temperature profiles available for the 2017 analysis as noted in Sect. 4.1. For both periods included in the comparison the agreement is better for night than day CALIOP measurements. It is noted that CALIOP day time measurements generally are more noisy than night time measurements. We also observe a lower standard deviation over ocean than over land. This is probably linked to the fact that retrievals over ocean are less uncertain: the surface emissivity

and temperature are more stable. In addition, the plume height is more constant over ocean (no local source), therefore less deviating from the a priori.

### 5.3 Qualitative profile comparison

In this section an effort is made to analyze the full MAPIR aerosol profiles. As the two previous sections already contain a comprehensive evaluation study on the AOD and aerosol layer mean altitude, here only a qualitative profile comparison is done.

Thus, we need data sets containing high-resolution aerosol profiles that are close to IASI data in both time and space. This small difference in time is very important, especially over source areas. We selected two ground-based lidar sites at relevant locations which offer aerosol extinction profiles with a very small time lag with IASI: M'Bour in Senegal and Al Dhaid in the United Arab Emirates (UAE), operated by the Laboratory of Atmospheric Optics (University of Lille, CNRS) and the Finnish Meteorological Institute respectively. The M'Bour site is situated at the coast of the Atlantic Ocean in the Sahel region,

where large amounts of dust are emitted yearly. The Al Dhaid site is located on the Arabian peninsula and also frequently experiences dust events. They are therefore both relevant locations to study the dust distribution. Additionally, we will explore data from the Cloud–Aerosol Transport System (CATS) onboard the International Space Station (ISS) to look for interesting profile comparisons. A description of these instruments was given in Sect. 2.

For each independent data set we co-locate the measurements with our IASI data in time and space. The matching criteria

however differ slightly in between data sets. For the ground-based lidar sites at M'Bour and Al Dhaid we select the IASI measurements that are within $0.5°$ of the station and within 1 hour of a lidar measurement. We compare the average of those MAPIR retrievals with the lidar profile averaged over a certain time period. At M'Bour, we take the average of all lidar profiles within an hour before the first IASI measurement and an hour after the last. At Al Dhaid we compare with a lidar profile that is averaged over 1 hour centered around the expected IASI overpass time. These differences in the temporal co-location arise

from the difference in the available data from the two stations. For the CATS data, we loop over all points on the ISS orbit in steps of $0.25°$. We compare the average CATS extinction profiles within $0.25°$ of those points with the average of MAPIR retrievals around the points, only if the time difference is less than 1 hour.

To account for the different resolution between the MAPIR and the various higher resolved lidar profiles, a smoothing is applied
to the regridded lidar profile $x_L$ by the MAPIR AK:

$$x'_L = x_a \exp\left( A \ln\left( \frac{x_L}{x_a} \right) \right), \tag{7}$$

where $x'_L$ is the smoothed or convolved lidar dust profile and $x_a$ and $A$ are the MAPIR a priori profile and AK. Equation (7) is based on the smoothing equation of Rodgers (2000) but transformed to suit our state vector. If the lidar measurement was the true atmospheric profile, then the smoothed lidar profile $x'_L$ represents how our observing system, the combination of the IASI
instrument and the MAPIR retrieval method, would retrieve it considering the limitations of the system (Rodgers and Connor, 2003). The lidar profiles both before and after smoothing will be presented.

### 5.3.1 M'Bour lidar

Some comparisons between the filtered extinction profiles from the M'Bour lidar and the retrieved MAPIR dust profiles can be found in Fig. 8 and Fig. 9. They cover the two-month period January–February 2015 and March–April 2016, respectively. Given
that both aerosol profiles are reported in different units and measured by other instruments, the results from this comparison should be treated with caution. For example in Fig. 8, if the colors representing the extinction at $532\,\mathrm{nm}$ differ from the colors representing particle density, this difference can be caused by the conversion factor used or by errors in either the lidar or MAPIR retrievals. It is more reliable to study the extent of the dust plumes in both data sets and verify if the occurrence of dust events is detected at the same time. This argument also applies for the analyses performed in the following subsections.
During winter 2015, M'Bour experiences dust events almost every day, as can be seen in the first plot of Fig. 8. These dust plumes roughly stretch between 0 and $2\,\mathrm{km}$ altitude, except at about mid-February where there is an elevated layer around 3 km. These features can be seen in the MAPIR profiles too. There is some amount of aerosols present at all co-located MAPIR retrievals in this period (see lowest plot in Fig. 8). The load is concentrated close to the surface with different intensities, the larger corresponding with the bigger events detected by the lidar. Moreover the higher dust layer around 17 February is also
detected by MAPIR. When comparing the smoothed lidar profiles (middle plot in Fig. 8) with the MAPIR retrievals, we come to similar conclusions. However, the intensity of dust events is probably sometimes underestimated by MAPIR. In January and February 2015, we see that in general MAPIR is good at detecting the dust events near M'Bour and additionally retrieves the vertical extent quite well.

Figure 9 shows the dust distribution near M'Bour in the spring of 2016. Again, mineral aerosols are detected on a daily basis,
with occasionally some larger events. In the middle and end of April, there are large dust plumes reaching an altitude of $4\,\mathrm{km}$ and $5\,\mathrm{km}$, respectively. Unfortunately, there are no co-located MAPIR profiles of good quality during the first event. Although a small part of it can still be seen in the MAPIR profile on 18 April, where it gets the same vertical extent as the lidar profile. The second event is better covered by MAPIR, where it reproduces the plume seen by the lidar adequately. The smoothed lidar

profiles agree well with the MAPIR profiles in April, but less in March. Especially on 19 March 2016, the averaged MAPIR profiles near M'Bour show a very high dust concentration around 2–3 km which is not seen in the lidar profiles. The reason for this rather contradictory result is not completely clear, but might be caused by a bad retrieval for that comparison. Among the averaged profiles, we observe a retrieval with an unrealistic surface temperature of more than 340 K, hence this difference is most probably due to a problem in the IASI data. Despite some differences in the comparisons, we believe that MAPIR observes the mineral dust profiles near M'Bour adequately, taking into account its limitations. This qualitative analysis of aerosol profiles at M'Bour supports our confidence in the value of the new MAPIR algorithm.

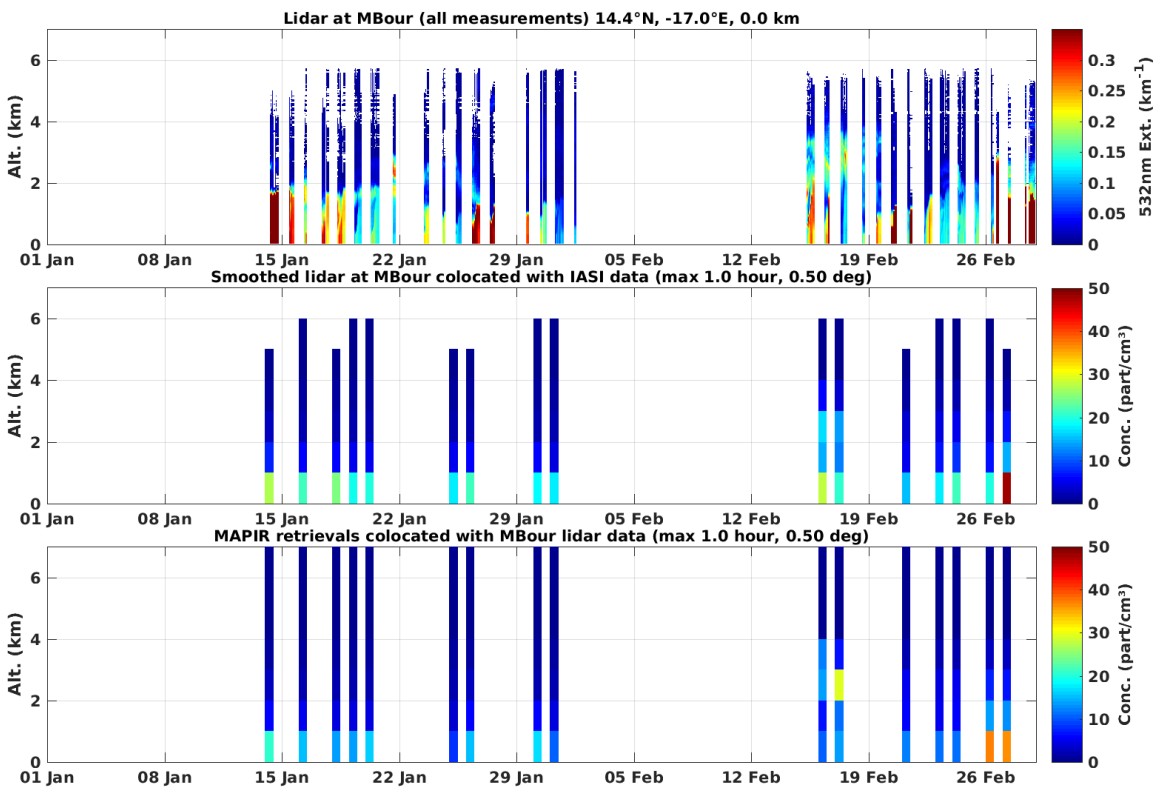

**Figure 8.** Mineral dust profile comparison at M'Bour site (Senegal) from 1 January to 1 March 2015. First plot gives the lidar data (extinction profiles at 532 nm) as provided for this study. On the second plot, the lidar data smoothed according to Eq. (7) is presented, for those times when there is a co-located MAPIR profile. The third plot presents the MAPIR profiles over time averaged around the M'Bour site.

### 5.3.2 Al Dhaid lidar

A comparison of the 2 months measurements (March to April 2018) at Al Dhaid with the MAPIR profiles is given Fig. 10. In general there are no large dust plumes detected by the lidar in March 2018. There is one event on 11 March where the lidar at

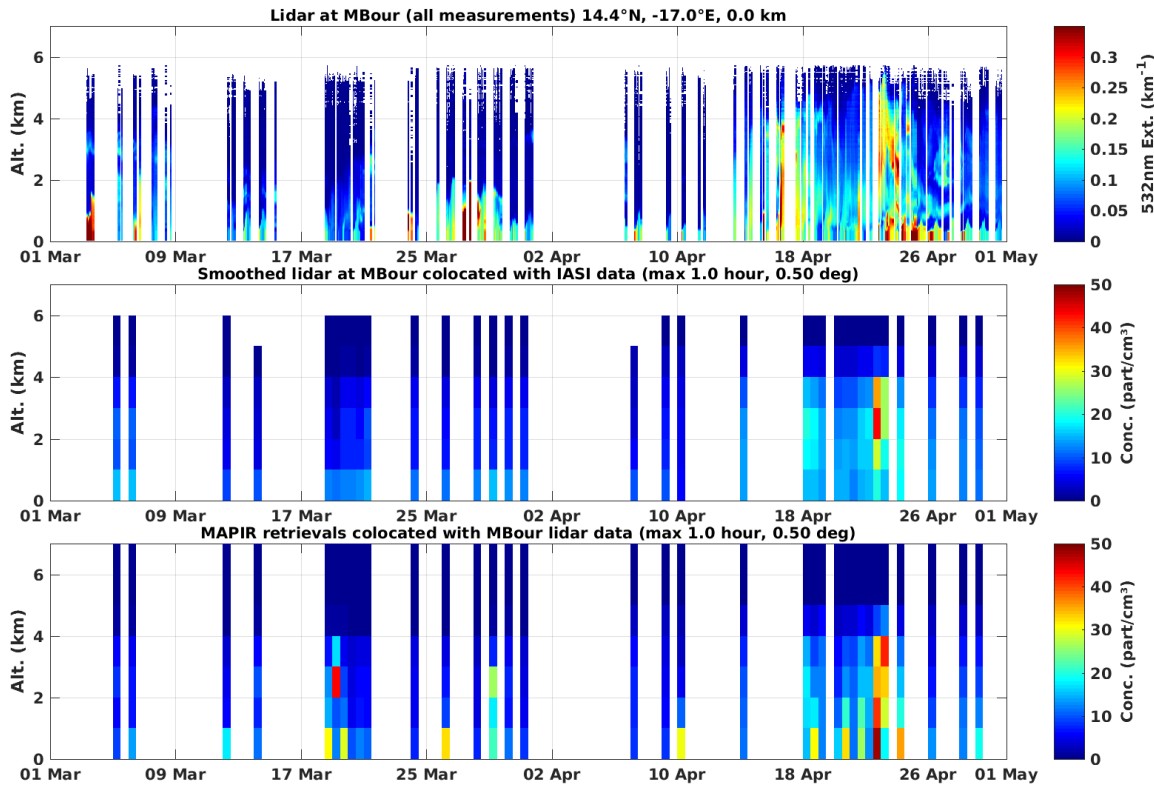

**Figure 9.** Same as Fig. 8 but for 1 March to 1 May 2016.

Al Dhaid detects a high aerosol load between 0 and 2 km altitude, but there is no co-located MAPIR profile of good quality to compare with. In the evening of 17 and 18 March, there is a faint elevated dust layer around 2 km that is seen by both the lidar and MAPIR. Likewise, the amount of aerosols concentrated below 1 km in the morning of 18 March is detected by both instruments. The second half of March does not contain any interesting events. However, the MAPIR profiles seem to have an almost continuous dust plume in the lowest layer, not as much detected by the lidar. The concentrations are relatively small and since the observing system has a low sensitivity in those cases, this background plume is probably linked to the a priori. It is also possible that the mean values of the LIDAR in the first layer are underestimated because the lower range limit of the lidar is about 180 m above ground level (Engelmann et al., 2016). Finally, the high aerosol concentration around 9 March as retrieved by MAPIR is probably the result of a bad retrieval. Since the retrieval passed all quality filters, it could also point to a problem in the ancillary data.

During April 2018, more interesting dust plumes pass nearby Al Dhaid. On 1 and 2 April, the lidar observes a dust layer reaching 2 and 4 km, respectively, which is similar to what MAPIR observes. However, MAPIR retrieves a much higher aerosol concentration near the surface. The dust plumes around 13 and 17 April are detected by both instruments, with similar

ranges. Another event occurs on 22 and 23 April. The lidar at Al Dhaid as well as the MAPIR retrievals close by, show larger dust signatures in that period. However, they do not completely agree on the altitudes of the layer.

Overall, this shows that MAPIR is reliable for the detection of mineral aerosols and even for the extent of the plumes. Based on the comparisons of this two-month period, yet a small overestimation of the lowest layer aerosol load near Al Dhaid appeared.

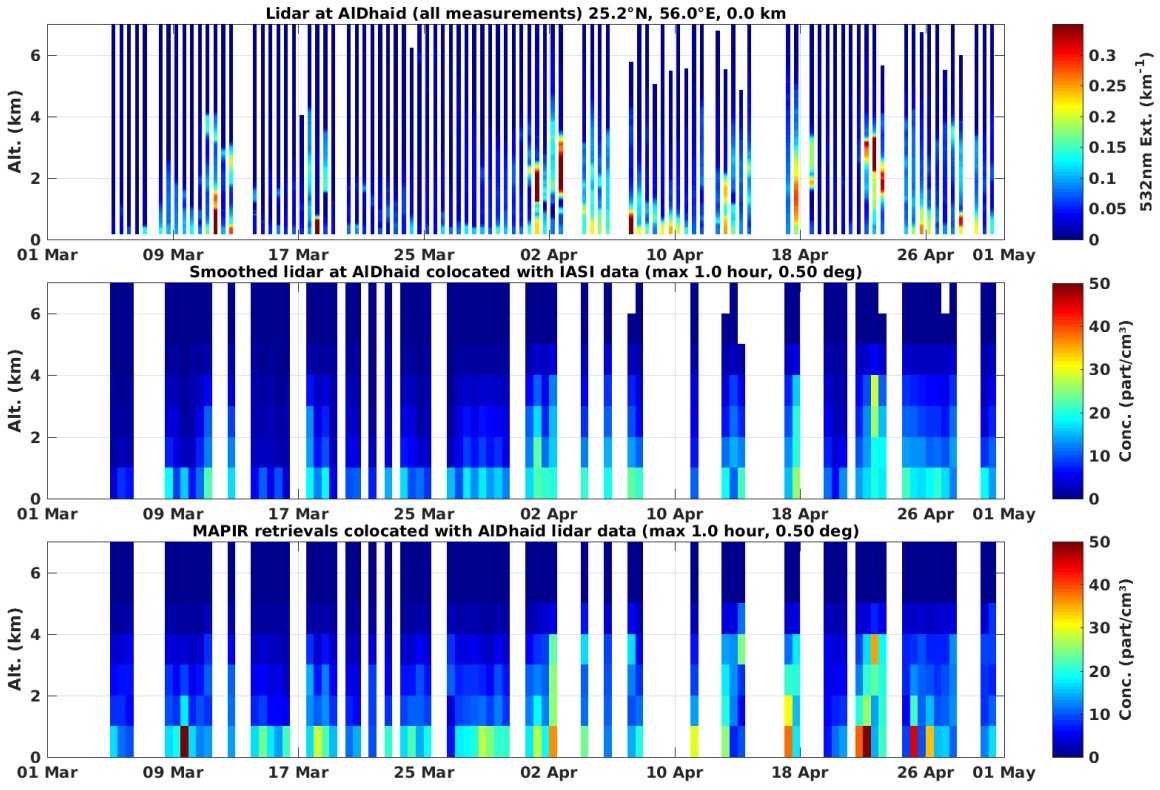

**Figure 10.** Mineral dust profile comparison at Al Dhaid site (UAE) from 1 March to 1 April 2018. First plot gives the lidar data (extinction profiles at 532 nm) as provided for this study. On the second plot, the lidar data smoothed according to Eq. (7) is presented, for those times when there is a co-located MAPIR profile. The third plot presents the MAPIR profiles over time averaged around the Al Dhaid site.

### 5.3.3  CATS

There were 1780 occurrences where CATS and IASI measurements could be co-located close both in time and space, not all of them containing interesting dust events. Two examples where high aerosol concentrations were observed are plotted in Fig. 11 and 12. They cover the Sahel on 16 February 2017 and Western Sahara on 19 June 2015, respectively.

In Figure 11 we see a spatially extended dust plume over the Sahel, with high concentrations relatively close to the surface. The width of the layer varies between 1 and 2 km, always reaching the ground. Similar features are observed by MAPIR: an

almost continuous, very dense surface layer of mineral aerosols along the track. The plume never reaches an altitude higher than 4 km. A bit further down the track, the dust plume is more elevated and spread out around 2–3 km height. Even though the load has decreased significantly, it is still detected by both CATS and IASI instruments.

Figure 12 presents another profile comparison of co-located CATS and IASI measurements. It shows several dust plumes over the Sahara in the evening of 19 June 2015. Both CATS and MAPIR retrieve a very dense plume extending from the surface to 5 km altitude around 6° E. Additionally, more westward two elevated layers around 5 km and 3–6 km can be observed by the two sensors. Since both CATS and MAPIR show such a good agreement, both in detection of dust events and extent of dust plumes, this is another example of the performance of MAPIR v4.1.

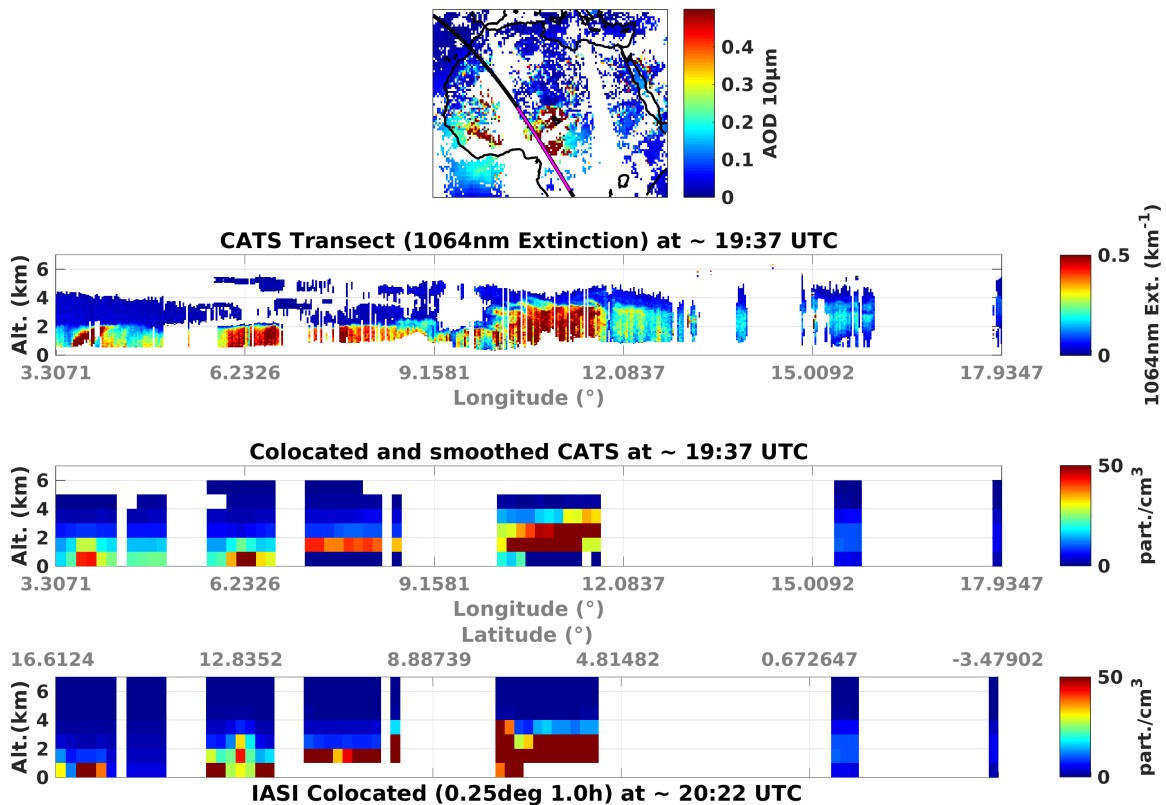

**Figure 11.** Mineral dust profile comparison along the CATS track on 16 February 2017. The first plot gives the global retrieved AOD by MAPIR together with the pathway of CATS that could be co-located with IASI in time and space. The part which corresponds to the plotted profiles below is given in pink. It covers the Sahel region. The second plot shows the dust extinction profiles along the pink track, as measured by CATS. On the third plot, the CATS data smoothed according to Eq. (7) is presented, on those locations where there is a MAPIR profile. Finally, the fourth plot presents the averaged MAPIR profiles along the track.

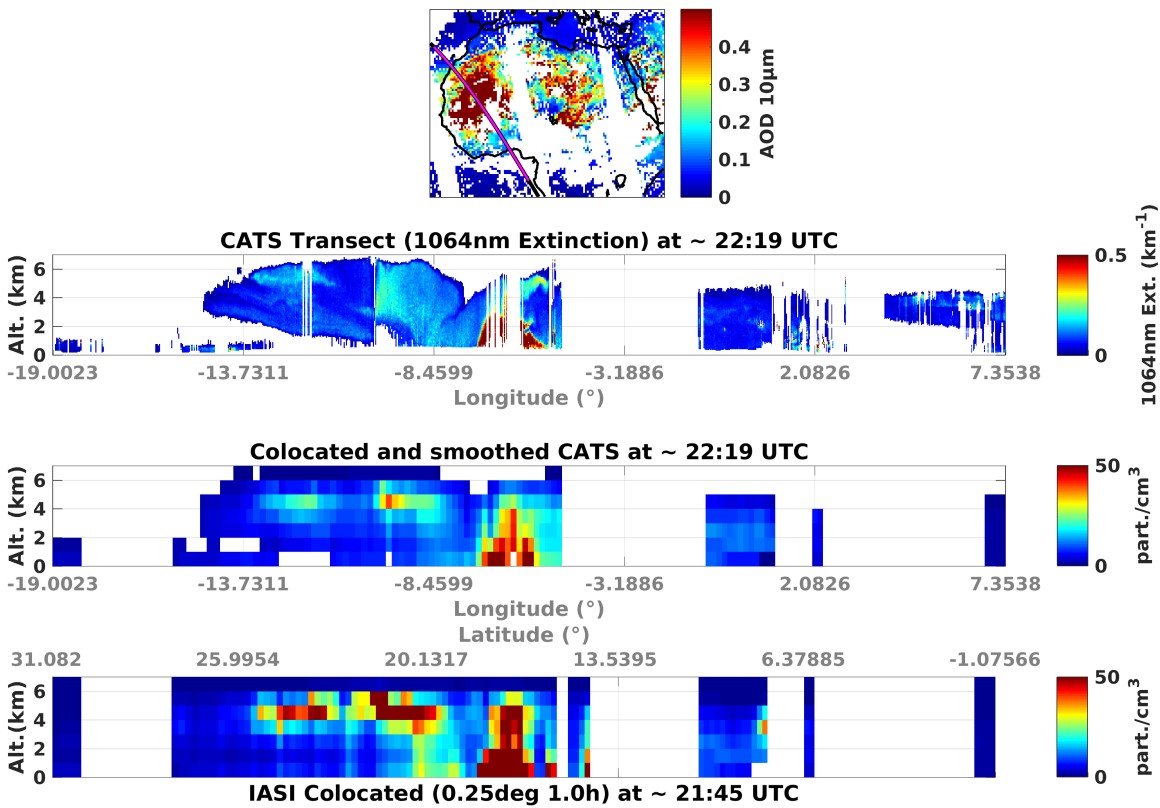

**Figure 12.** Same as Fig. 11 but with a co-located track on 19 June 2015. The pink track covers Western Sahara.

## 6   Discussion, conclusion and further work

In this work, we describe and provide an evaluation of the updated Mineral Aerosol Profiling from Infrared Radiances (MAPIR) algorithm version 4.1, retrieving dust aerosol concentration profiles in seven 1 km-thick layers centered at 0.5 to 6.5 km altitude, using the optimal estimation method applied on thermal infrared radiances measured by the Infrared Atmospheric
5   Sounding Interferometer onboard the MetOp satellite series. The new version of MAPIR was developed to cope with known issues of earlier versions: the high fraction of bad retrievals over Sahara and Sahel regions (about 40 % on average), the huge overestimation of the aerosol optical depth (AOD, overestimated on average by 0.28) and the large computation time. The main modifications to the algorithm are: (1) a faster radiative transfer (RT) model Radiative Transfer for TOVS (RTTOV) to replace LIDORT, (2) using the logarithmic concentrations in the retrieval to avoid numerically plausible but nonphysical nega-
10   tive concentrations and (3) adding the Levenberg–Marquardt modification of the OEM for a better and faster convergence. All input parameters, such as the IASI level 1 spectra, aerosol properties, temperature and humidity profiles and other ancillary data remain unchanged with regard to the previous MAPIR versions.

Using concentrations in the logarithmic space induces different underlying constraints on the state vector than before. In cases with high aerosol concentrations, the retrievals will be less constrained by the a priori and more sensitive to the true profile. Conversely, retrievals are more constrained in regions with low aerosol concentrations (Deeter et al., 2007).

MAPIR v4.1 has been applied to almost 11 years of IASI measurements, resulting in a large data set that makes it possible to accurately assess the quality of the updated algorithm. The results show a significant increase in retrieval quality (from 40 % to 16 % bad retrievals) and convergence (from about 0.8 % to 0.6 % non-converging). There is an increase of MAPIR data quality over time, most likely due to the evolution of the different EUMETSAT IASI level 2 products for temperature profiles used in MAPIR retrievals. The goodness of fit of the retrievals (after quality filtering) is represented by a median root mean square of the spectral residuals of 0.32 K.

The information content of the retrievals is assessed through the so-called averaging kernels (AKs) obtained from the OEM. The trace of those AKs provides the number of degrees of freedom (DOF) or independent pieces of information which can be retrieved from the observations, considering the instrumental noise and the a priori knowledge of the atmosphere. For dusty scenes (AOD $\geq 0.5$) there is a median DOF of 1.4. For non-dusty scenes, the DOF can be very low due to the constraints associated with log-normal retrievals.

This new 3-D data set of mineral dust has been evaluated using data from the ground-based AERONET network, the CALIOP satellite data, data from the CATS instrument onboard the international space station and data from two ground-based lidar sites, at M'Bour (Senegal) and Al Dhaid (United Arab Emirates).

First, a selection of 72 AERONET sites was used to compare the dust AOD obtained from the integrated MAPIR profiles to the AERONET SDA coarse mode AOD at 500 nm. Overall there is a strong correlation of up to 0.88, especially over Northern India and Sahara and Sahel regions. A limited number of stations show a weaker correlation which can be caused by various reasons: a specific station location between different air masses, biased statistics due to misrepresentation of the actual AOD distribution at a specific station, AERONET SDA being sensitive to another type of coarse mode aerosol than dust or unrealistic MAPIR retrieval input data (temperature profiles, surface emissivity) leading to lesser quality retrieved information. However, in general MAPIR is quite good at reproducing AERONET AOD with a mean positive AOD bias of only 0.04 over all stations along the whole time series. The AOD overestimation observed with previous versions of MAPIR is therefore now solved.

The MAPIR mean dust layer altitudes were compared with the CALIOP geometric mean and cumulative extinction dust layer altitudes. In those comparisons, the time difference between IASI and CALIOP (3 to 5 hours) is accounted for, using the FLEXTRA transport model to simulate the transport of the air masses observed by CALIOP backwards in time to the IASI observation time. MAPIR v4.1 underestimates the CALIOP cumulative extinction and geometric mean dust layer altitudes for the 2017 sample by $0.322 \pm 1.044$ km and $0.640 \pm 1.031$ km, respectively. Considering that the MAPIR profiles are retrieved with a resolution of 1 km, this comparison shows the dust layer altitude from MAPIR v4.1 is rather accurate. The standard deviation of the difference between the MAPIR and CALIOP altitude is consistently smaller by about 0.1 km for MAPIR v4.1 compared with earlier MAPIR versions. Furthermore, comparing 2010 and 2017 results, the improved IASI temperature profiles (from EUMETSAT), used as input to our retrievals, appears to lead to smaller differences between MAPIR and CALIOP altitudes.

Finally, the full vertical profiles were qualitatively compared with data from two ground-based lidar sites and from the CATS instrument. Four months of lidar measurements at M'Bour near Dakar, Senegal, were compared with the associated MAPIR profiles. Both instruments detected similar dust plumes at the same times. In Al Dhaid, United Arab Emirates, almost all dust events that were detected by the lidar during the two-month comparison period were also seen in the MAPIR data. However,

MAPIR also detects a constant low altitude low concentration dust layer not seen with the lidar. A very good agreement was obtained when comparing the MAPIR profiles with the measured extinction by CATS. MAPIR showed the ability to reproduce the CATS dust plumes both at low and high altitudes over bright surfaces, such as Sahara and Sahel. Overall, these qualitative profile comparisons give us confidence in the competence of MAPIR to retrieve mineral aerosol profiles. In particular, the full profile comparisons were selected as being in areas close to sources, where the temporal difference with CALIOP does not

ensure that both instruments observe the same air mass, while with CATS and the ground-based instruments a maximum time difference of 1 hour was accepted for the comparisons.

We have shown that the new MAPIR algorithm provides reliable AOD, dust layer mean altitude and profiles. Together with the extensive spatial and temporal coverage of IASI, MAPIR v4.1 is a new powerful tool to improve the understanding of the 3-D dust distribution over time.

Future work to further improve the MAPIR algorithm can include the better characterization of the dust aerosols. Possible improvements are the use of more recent refractive index data, for example those of Di Biagio et al. (2017) and the use of a bi-modal particle size distribution. Both represent significant scientific work: the development of an automated selection of the best refractive index and/or the retrieval of an additional parameter being the ratio between the 2 modes of the particle size distribution. In addition, assuming non-spherical particles would make the aerosol representation more realistic, which is

especially important for the conversion to visible AOD. Further, the product would benefit from a better cloud and dust filter. An improved cloud filter would add valuable information on the most intense dust events as those are often missed now, being misflagged as clouds in the IASI operational level 2 cloud product. Finally, as the retrieval is much affected by the quality of surface emissivity and temperature profiles, improved data sets of these input parameters could also increase the accuracy of MAPIR in the future.

*Data availability.*  Under the Copernicus Climate Change Service aerosols project, the MAPIR dust 10 µm and 550 nm AOD and the MAPIR dust aerosol mean altitude were submitted to the Copernicus Climate Data Store where they currently undergo technical processing. The full profiles (and the AOD and mean altitude) from MAPIR are available upon request to the authors

## Appendix A: AERONET - MAPIR data

This appendix contains additional data from the comparison between AOD at AERONET stations and AOD from MAPIR. In Table A1 a list of the AERONET stations that were used for this study is given together with their coordinates and correlation parameters.

Table A1: List of the 72 AERONET sites selected for the evaluation study, together with their geographical coordinates and the results from the regression analysis: geographical location, latitude, longitude, number of observations used in the analysis, Pearson correlation coefficient between AOD at the site and MAPIR, slope and Y-intersection of the regression line. The standard deviation of the correlation and regression parameters is also given. Coastal sites where the AERONET coarse mode AOD is potentially impacted by sea salt are marked with an asterisk after their name.

| Site | Geogr. terr. | Lat.($°$) | Long.($°$) | Nb | Corr. | $\sigma_{corr}$ | Slope | $\sigma_{slope}$ | Inters. | $\sigma_{inters}$ |
|---|---|---|---|---|---|---|---|---|---|---|
| Abu Al Bukhoosh* | UAE | 25.50 | 53.15 | 355 | 0.73 | 0.04 | 0.48 | 0.02 | 0.14 | 0.01 |
| Alboran* | Spain | 35.94 | −3.35 | 191 | 0.71 | 0.05 | 0.56 | 0.04 | 0.05 | 0.01 |
| Arica* | Chile | −18.47 | −70.31 | 493 | 0.07 | 0.05 | 0.03 | 0.02 | 0.03 | 0.00 |
| Bakersfield | USA | 35.33 | −119.00 | 715 | 0.22 | 0.04 | 0.68 | 0.12 | 0.04 | 0.01 |
| Bambey-ISRA | Senegal | 14.71 | −16.48 | 157 | 0.75 | 0.05 | 0.86 | 0.06 | −0.02 | 0.03 |
| Barbados | Barbados | 13.15 | −59.62 | 133 | 0.66 | 0.07 | 0.51 | 0.05 | 0.06 | 0.01 |
| SALTRACE* | | | | | | | | | | |
| Beijing-CAMS | China | 39.93 | 116.32 | 1562 | 0.64 | 0.02 | 0.89 | 0.03 | 0.12 | 0.00 |
| Beijing | China | 39.98 | 116.38 | 833 | 0.71 | 0.02 | 0.75 | 0.03 | 0.14 | 0.00 |
| Beijing RADI | China | 40.00 | 116.38 | 172 | 0.85 | 0.04 | 1.09 | 0.05 | 0.12 | 0.01 |
| Ben Salem | Tunisia | 35.55 | 9.91 | 514 | 0.79 | 0.03 | 0.80 | 0.03 | 0.06 | 0.00 |
| Blida | Algeria | 36.51 | 2.88 | 800 | 0.73 | 0.02 | 0.70 | 0.02 | 0.05 | 0.00 |
| Cairo EMA 2 | Egypt | 30.08 | 31.29 | 2022 | 0.66 | 0.02 | 0.60 | 0.02 | 0.03 | 0.00 |
| Calhau* | Cape Verde | 16.86 | −24.87 | 391 | 0.75 | 0.03 | 0.65 | 0.03 | 0.09 | 0.01 |
| Camaguey* | Cuba | 21.42 | −77.85 | 1347 | 0.54 | 0.02 | 0.52 | 0.02 | 0.00 | 0.00 |
| Cape San Juan* | Puerto Rico | 18.38 | −65.62 | 1052 | 0.71 | 0.02 | 0.53 | 0.02 | 0.02 | 0.00 |
| Capo Verde* | Cape verde | 16.73 | −22.94 | 177 | 0.75 | 0.05 | 0.64 | 0.04 | 0.09 | 0.02 |
| CUT-TEPAK* | Cyprus | 34.67 | 33.04 | 1206 | 0.61 | 0.02 | 0.46 | 0.02 | 0.04 | 0.00 |
| Dakar* | Senegal | 14.39 | −16.96 | 2410 | 0.57 | 0.02 | 0.62 | 0.02 | 0.13 | 0.01 |
| Dhadnah* | UAE | 25.51 | 56.32 | 496 | 0.74 | 0.03 | 0.74 | 0.03 | 0.10 | 0.01 |
| Dhaka University | Bangladesh | 23.73 | 90.40 | 794 | 0.77 | 0.02 | 0.76 | 0.02 | 0.01 | 0.00 |
| Dushanbe | Tajikistan | 38.55 | 68.86 | 2084 | 0.74 | 0.02 | 0.74 | 0.02 | 0.04 | 0.00 |
| Eilat* | Israel | 29.50 | 34.92 | 2243 | 0.51 | 0.02 | 0.55 | 0.02 | 0.07 | 0.00 |
| Gandhi College | India | 25.87 | 84.13 | 1377 | 0.74 | 0.02 | 0.87 | 0.02 | 0.09 | 0.01 |
| Guadeloup* | France/Carribean | 16.22 | −61.53 | 903 | 0.73 | 0.02 | 0.57 | 0.02 | 0.01 | 0.00 |
| Gual Pahari | India | 28.43 | 77.15 | 473 | 0.86 | 0.02 | 0.92 | 0.03 | 0.08 | 0.01 |
| Hada El-Sham | Saudi Arabia | 21.80 | 39.73 | 469 | 0.60 | 0.04 | 0.58 | 0.04 | 0.21 | 0.01 |
| ICIPE-Mbita | Kenia | −0.43 | 34.21 | 826 | 0.54 | 0.03 | 0.47 | 0.03 | 0.02 | 0.00 |

*Continued on next page*

Table A1: List of the 72 AERONET sites selected for the evaluation study, together with their geographical coordinates and the results from the regression analysis: geographical location, latitude, longitude, number of observations used in the analysis, Pearson correlation coefficient between AOD at the site and MAPIR, slope and Y-intersection of the regression line. The standard deviation of the correlation and regression parameters is also given. Coastal sites where the AERONET coarse mode AOD is potentially impacted by sea salt are marked with an asterisk after their name.

| Site | Geogr. terr. | Lat.(°) | Long.(°) | Nb | Corr. | $\sigma_{corr}$ | Slope | $\sigma_{slope}$ | Inters. | $\sigma_{inters}$ |
|---|---|---|---|---|---|---|---|---|---|---|
| IER Cinzana | Mali | 13.28 | −5.93 | 114 | 0.84 | 0.05 | 0.79 | 0.05 | 0.08 | 0.02 |
| Ilorin | Nigeria | 8.48 | 4.67 | 1340 | 0.85 | 0.01 | 0.71 | 0.01 | 0.13 | 0.01 |
| IMS-METU-ERDEMLI* | Turkey | 36.56 | 34.26 | 2498 | 0.27 | 0.02 | 0.15 | 0.01 | 0.06 | 0.00 |
| Jaipur | India | 26.91 | 75.81 | 2651 | 0.81 | 0.01 | 1.00 | 0.01 | 0.11 | 0.00 |
| Kanpur | India | 26.51 | 80.23 | 2751 | 0.81 | 0.01 | 0.86 | 0.01 | 0.10 | 0.00 |
| Karachi | Pakistan | 24.95 | 67.14 | 1344 | 0.76 | 0.02 | 0.74 | 0.02 | 0.08 | 0.01 |
| Kathmandu-Bode | Nepal | 27.68 | 85.39 | 505 | 0.79 | 0.02 | 0.92 | 0.02 | −0.00 | 0.00 |
| KAUST Campus* | Saudi Arabia | 22.30 | 39.10 | 1550 | 0.67 | 0.02 | 0.56 | 0.02 | 0.12 | 0.00 |
| Koforidua ANUC | Ghana | 6.11 | −0.30 | 404 | 0.86 | 0.03 | 0.65 | 0.02 | 0.11 | 0.01 |
| Kuwait University | Kuwait | 29.32 | 47.97 | 299 | 0.19 | 0.06 | 0.47 | 0.14 | 0.34 | 0.06 |
| Lahore | Pakistan | 31.48 | 74.26 | 1254 | 0.72 | 0.02 | 0.86 | 0.02 | 0.15 | 0.01 |
| La Laguna* | Tenerife | 28.48 | −16.32 | 1621 | 0.69 | 0.02 | 0.68 | 0.02 | 0.04 | 0.00 |
| Lampedusa* | Italy | 35.52 | 12.63 | 1690 | 0.46 | 0.02 | 0.31 | 0.02 | 0.07 | 0.00 |
| La Parguera* | Puerto Rico | 17.97 | −67.05 | 2679 | 0.72 | 0.01 | 0.61 | 0.01 | 0.01 | 0.00 |
| Lumbini | Nepal | 27.49 | 83.28 | 582 | 0.82 | 0.02 | 0.94 | 0.03 | 0.09 | 0.01 |
| Mandalay MTU | Myanmar | 21.97 | 96.19 | 383 | 0.73 | 0.04 | 0.75 | 0.04 | 0.02 | 0.00 |
| Masdar Institute* | UAE | 24.44 | 54.62 | 1530 | 0.79 | 0.02 | 0.80 | 0.02 | 0.09 | 0.00 |
| MCO-Hanimaadhoo* | Maldives | 6.78 | 73.18 | 1362 | 0.62 | 0.02 | 0.45 | 0.02 | 0.02 | 0.00 |
| Mezaira | UAE | 23.10 | 53.75 | 1905 | 0.48 | 0.02 | 0.75 | 0.03 | 0.12 | 0.01 |
| Mussafa* | UAE | 24.37 | 54.47 | 563 | 0.82 | 0.02 | 0.64 | 0.02 | 0.09 | 0.01 |
| Myanmar | Myanmar | 16.86 | 96.15 | 154 | 0.54 | 0.07 | 0.45 | 0.06 | 0.02 | 0.00 |
| Nainital | India | 29.36 | 79.46 | 361 | 0.77 | 0.03 | 1.27 | 0.06 | 0.03 | 0.01 |
| NEON GUAN* | Puerto Rico | 17.97 | −66.87 | 197 | 0.58 | 0.06 | 0.50 | 0.05 | 0.03 | 0.00 |
| Nes Ziona* | Israel | 31.92 | 34.79 | 1675 | 0.63 | 0.02 | 0.54 | 0.02 | 0.07 | 0.00 |
| New Delhi IMD | India | 28.59 | 77.22 | 168 | 0.82 | 0.04 | 1.00 | 0.05 | 0.08 | 0.02 |
| New Delhi | India | 28.63 | 77.18 | 134 | 0.76 | 0.06 | 0.71 | 0.05 | 0.15 | 0.02 |
| NUIST | China | 32.21 | 118.72 | 182 | 0.52 | 0.06 | 0.55 | 0.07 | 0.11 | 0.02 |
| Oujda | Morroco | 34.65 | −1.90 | 1438 | 0.88 | 0.01 | 0.89 | 0.01 | 0.04 | 0.00 |
| Pantnagar | India | 29.05 | 79.52 | 318 | 0.71 | 0.04 | 0.82 | 0.05 | 0.09 | 0.01 |
| Ragged Point* | Barbados | 13.17 | −59.43 | 2271 | 0.66 | 0.02 | 0.55 | 0.01 | 0.02 | 0.00 |
| Saada | Morocco | 31.63 | −8.16 | 364 | 0.79 | 0.03 | 0.95 | 0.04 | 0.05 | 0.01 |

*Continued on next page*

Table A1: List of the 72 AERONET sites selected for the evaluation study, together with their geographical coordinates and the results from the regression analysis: geographical location, latitude, longitude, number of observations used in the analysis, Pearson correlation coefficient between AOD at the site and MAPIR, slope and Y-intersection of the regression line. The standard deviation of the correlation and regression parameters is also given. Coastal sites where the AERONET coarse mode AOD is potentially impacted by sea salt are marked with an asterisk after their name.

| Site | Geogr. terr. | Lat.(°) | Long.(°) | Nb | Corr. | $\sigma_{corr}$ | Slope | $\sigma_{slope}$ | Inters. | $\sigma_{inters}$ |
|------|-------------|---------|----------|-----|-------|-----------------|-------|------------------|---------|-------------------|
| SACOL | China | 35.95 | 104.14 | 410 | 0.94 | 0.02 | 0.99 | 0.02 | 0.04 | 0.01 |
| Santa Cruz Tenerife* | Tenerife | 28.47 | −16.25 | 2620 | 0.57 | 0.02 | 0.53 | 0.02 | 0.05 | 0.00 |
| Sao Tome* | Sao Tome and Principe | 0.37 | 6.71 | 96 | 0.86 | 0.05 | 0.87 | 0.05 | 0.00 | 0.01 |
| SEDE BOKER | Israel | 30.86 | 34.78 | 3525 | 0.64 | 0.01 | 0.62 | 0.01 | 0.05 | 0.00 |
| Shagaya Park | Kuwait | 29.21 | 47.06 | 156 | 0.65 | 0.06 | 0.46 | 0.04 | 0.02 | 0.01 |
| Solar Village | Saudi Arabia | 24.91 | 46.40 | 1996 | 0.57 | 0.02 | 0.52 | 0.02 | 0.11 | 0.01 |
| Taihu | China | 31.42 | 120.22 | 361 | 0.64 | 0.04 | 0.80 | 0.05 | 0.04 | 0.01 |
| Tamanrasset INM | Algeria | 22.79 | 5.53 | 3187 | 0.41 | 0.02 | 1.04 | 0.04 | 0.41 | 0.01 |
| Tizi Ouzou | Algeria | 36.70 | 4.06 | 1177 | 0.70 | 0.02 | 0.68 | 0.02 | 0.06 | 0.00 |
| Tunis Carthage | Tunisia | 36.84 | 10.20 | 1196 | 0.70 | 0.02 | 0.57 | 0.02 | 0.05 | 0.00 |
| UPC–GEAB– Valledupar | Colombia | 9.56 | −73.33 | 111 | −0.11 | 0.09 | −0.07 | 0.07 | 0.04 | 0.00 |
| Weizmann Institute* | Israel | 31.91 | 34.81 | 944 | 0.71 | 0.02 | 0.61 | 0.02 | 0.06 | 0.00 |
| XiangHe | China | 39.75 | 116.96 | 2744 | 0.60 | 0.02 | 0.70 | 0.02 | 0.13 | 0.00 |
| Zinder Airport | Niger | 13.78 | 8.99 | 203 | 0.49 | 0.06 | 0.70 | 0.09 | 0.24 | 0.04 |

*Competing interests.* The authors declare that they have no conflict of interest.

*Acknowledgements.* The development of the MAPIR algorithm was supported by the European Space Agency (ESA) as part of the Aerosol_cci project, by the Belgian Science Policy (BELSPO) supplementary researcher program and by the ESA/BELSPO PRODEX program under the IASI.flow project phases 2 and 3. The data processing was partially funded by the European Center for Medium-Range Weather Forecast (ECMWF) Copernicus Climate Change Services (C3S) project 312a and 312b. We would like to thank the EUMETSAT service for providing the IASI data. Our thanks goes also to all involved with the necessary data sets for input and evaluation. We thank the principal investigators of AERONET and their staff for establishing and maintaining the 72 sites used in the AOD analysis study. We are grateful to the NASA Atmospheric Science Data Center for providing CALIOP data. We would like to acknowledge ACTRIS-France, the Service

National d'Observations PHOTONS/AERONET and the IRD (Institut de Recherche pour le Développement) for the lidar data near Dakar. The lidar profiles used for the profile comparisons at Al Dhaid, UAE are based on work supported by the National Center of Meteorology, Abu Dhabi, UAE, under the UAE Research Program for Rain Enhancement Science. Our thanks goes to Matthew McGill, John Yorks and Dennis Hlavka for establishing the CATS data set. We thank Jeffrey Reid, Logan Lee and Jianglong Zhang for interesting and useful scientific exchanges about CATS. The participants of the COST action CA16202 inDust are praised for fruitful discussions. We also acknowledge the BIRA-IASB IT team for providing support in the processing of the MAPIR data set, and Jerôme Vidot (Meteo France) for his help in starting up with RTTOV and preparing the aerosols table for our specific parameters.

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
