# Peer review of "The Mineral Aerosol Profiling from Infrared Radiances (MAPIR) algorithm: version 4.1 description and evaluation"

_Atmospheric Measurement Techniques, 2019_

## Referee Comment (RC1) · Anonymous Referee #1 · 29 Mar 2019

Review of the manuscript entitled '' The Mineral Aerosol Profiling from Infrared Radiances (MAPIR) algorithm: version 4.1 description and validation" by Sieglinde Callewaert et al.

This manuscript describes a new method applied to an algorithm that retrieves mineral aerosol profiles from infrared radiances. The method can retrieve the 3D structure of the dust fields which permits to access both aerosol optical depth (AOD) and the altitude at which the dust layer is located. The manuscript first describes the algorithm and then shows examples of extended retrievals for scenes in June 2018. A detailed comparison is then made with AOD 4 AERONET stations during a multi-year period.

[Figure]

The comparison proceeds with an estimation of the height of the transported dust aerosol layer and a comparison with LIDAR profiles over M'Bour, Senegal and Al Dhaid In the United Arab Emirates. The presentation of the paper needs to be improved and I have several comments that need to be addressed before this paper can be accepted for publication to AMT. Please note that I do not have the background to judge the relevance of the choices made in the algorithm, hence I will focus my comments in relation to the dust cycle.

Main comments:

The authors indicate the existence of parallel efforts such as the ones described in Clarisse et al. 2019 and by Cuesta et al., 2015 to make mineral aerosol retrieval. Although these efforts are mentioned in the introduction there is no mention to the reader of how the presented work here differ by its methods upon these two other studies. I am not asking for quantitative statements but rather that the readers be made aware of the limitations of each of the studies.

A lot of effort is devoted to improvements in the algorithm so that the screening of the scene is improved and the detectability of dust over desert areas with low emissivity becomes feasible. I was surprised that the dust properties chosen in the retrieval were not looked at critically. With regards to either the particle size distribution, and the refractive indices in the shortwave taken from Volz (1972, 1973) and Shettle and Fenn (1979) there are many studies showing large deficiencies in these description of dust properties. A recent study from Ryder et al. (2018) where the full size distribution of dust is measured clearly indicates that a single mode with an effective radius of 2um and a width of 2.0 accounts only for a small part of this size distribution. Have the authors tried to estimate the influence this could have on their retrievals? The following references Dubovik et al. (2002); Sinyuk et al. (2003); Colarco et al., (2003); Balkanski et al. (2007) and Di Biagio et al. (2019) all show that the refractive indices used here are much too absorbing compared to any dust sample examined over the last 10 years. The same question than above should be addressed by the authors: how does

a correction in the refractive index use would influence their results and the description of the 3D dust distributions that they provide.

When examining the AERONET coarse mode AOD, I did not see any discussion about the possible influence of seasalt on these AOD. This would be particularly sensitive for marine or coastal sites near sea-level and could explain a good part of the discrepancies at these sites. Please indicate it, or try to estimate how much the total coarse mode AOD could deviate from the dust coarse mode AOD at these sites.

I propose a change in the structure of the text of paragraph 5.3. The description of the Lidar characteristics for the 3 lidars at M'Bour, at Al Dhaid and on-board the space station should have been given on an earlier part of the paper so that the authors focus only on the comparison which is the title of paragraph 5.3. This whole paragraph needs to be better organized and better written if you want to keep the attention of the reader. This paper is relatively long so this part has to be well written.

And last but not least of the major comments, that Data availability statement at the bottom of page 28 is very fuzzy as it is. Since this project is financed by Copernicus, the data availability is mandatory and cannot be delayed in time. How can someone interested in studying this dataset access it?

Minor Comments:

In the introduction, the authors should mention that 3D fields of dust based upon observations are described in Ridley et al. (2016).

Page 6, line 16: please delete the sentence: '' Above 7 km there is rarely found mineral dust particles, as is shown by a CALIOP based 3 -D climatology described in Winker et al. (2013)." This statement is inaccurate, many lidar profiles above the Mediterranean Sea show dust plumes above 7 and even 10 km. Hence, saying that dust is rarely seen above 7 km can mislead readers.

Page 16, line 3: you mention that you chose 4 surface stations to conduct a more

thorough comparison with AERONET, please indicate how these stations were chosen.

Page 16, line 18, please change: Indeed, for the a priori, a monthly climatology over 8 years is used (...) To A monthly climatology over 8 years is used for the a priori (...)

Page 16, line 31-32: '' The first period is identical to the same time and region used by Kylling et al. (2018): 18..", please indicate the lat, lon of the region that you mention here.

Page 16; in paragraph 5.2 you should indicate that if the dust layer is situated above 7km, it will be missed by your algorithm.

Page 21 lines 19-20: stating that '' This qualitative analysis of aerosol profiles at M'Bour supports our confidence in the value of the new MAPIR algorithm." Is not justified since we do not have, as of today, a golden standard for dust profiles to judge when we can be confident in a dust retrieving algorithm. Please delete this sentence.

Page 25 line 14, the work you do here is more an evaluation than a validation since you would need very well defined uncertainties on the dust quantities that are measured to make that validation. I propose that you change the term 'validation' to 'evaluation in the title of this manuscript and change the text 'provide validation' to 'evaluate' in this sentence.

Page 27, line 12, there is a typo that your co-authors should have picked up: the units of extinction should be km-1 and not km.

Page 28, line 8, there is one comma too many, please delete the comma and change the text from: '' In Al Dhaid, United Arab Emirates, almost all dust events that were detected by the lidar during the two-month comparison period, were also seen in the MAPIR data." To '' In Al Dhaid, United Arab Emirates, almost all dust events that were detected by the lidar during the two-month comparison period were also seen in the MAPIR data."

References

Dubovik, O., Holben, B., Eck, T., Smirnov, A., Kaufman, Y., King, M., Tanré, D., and Slutsker, I.: Variability of absorption and optical properties of key aerosol types observed in world-wide locations, J. Atmos. Sci., 59, 590–608, 2002.

Sinyuk, A., O. Torres, and O. Dubovik (2003), Combined use of satellite and surface observations to infer the imaginary part of the refractive index of Saharan dust, Geophys. Res. Lett., 30(2), 1081, doi:10.1029/2002GL016189.

Colarco, P. R., O. B. Toon, and B. N. Holben (2003), Saharan dust transport to the Caribbean during PRIDE: 1. Influence of dust sources and removal mechanisms on the timing and magnitude of downwind aerosol optical depth events from simulations of in situ and remote sensing observations, J. Geophys. Res., 108(D19), 8589, doi:10.1029/2002JD002658.

Balkanski, Y., Schulz, M., Claquin, T., and Guibert, S.: Reevaluation of Mineral aerosol radiative forcings suggests a better agreement with satellite and AERONET data, Atmos. Chem. Phys., 7, 81-95, https://doi.org/10.5194/acp-7-81-2007, 2007.

Di Biagio, C., Formenti, P., Balkanski, Y., Caponi, L., Cazaunau, M., Pangui, E., Journet, E., Nowak, S., Andreae, M. O., Kandler, K., Saeed, T., Piketh, S., Seibert, D., Williams, E., and Doussin, J.-F.: Complex refractive indices and single scattering albedo of global dust aerosols in the shortwave spectrum and relationship to iron content and size, Atmos. Chem. Phys. Discuss., https://doi.org/10.5194/acp-2019-145, in review, 2019.

Ridley, D. A., Heald, C. L., Kok, J. F., and Zhao, C.: An observationally constrained estimate of global dust aerosol optical depth, Atmos. Chem. Phys., 16, 15097-15117, https://doi.org/10.5194/acp-16-15097-2016, 2016.

---

## Referee Comment (RC2) · Anonymous Referee #2 · 1 Apr 2019

The authors present a useful multi-sensor study, a fairly comprehensive validation of the updated version 4.1 IASI MAPIR dust retrieval algorithm with respect to data from numerous AERONET sites, CALIOP spaceborne lidar data, ground-based lidar data, and lidar data from the CATS instrument onboard the International Space Station. Updates to the previous version of the algorithm include a change of the radiative transfer model used (RTTOV), a change to logarithmic aerosol concentrations, and the use of the Levenberg-Marquardt modification of the Gauss-Newton iteration scheme. The updates seem to enhance the performance of the MAPIR AOD retrieval algorithm with respect to AERONET measurements, reducing the previous bias of +0.28 to -0.04 in the new version. Validation with various lidar instruments indicates that the MAPIR

algorithm has some useful skill in retrieving the dust aerosol height.

One broader question that I have, which I feel is considered implicitly but not explicitly throughout the manuscript, is whether there is much to be said about the relationship between the IASI-derived atmospheric profiles of water vapour (and temperature) and the dust aerosol profiles (validated in Sections 5.2 and 5.3). In the infrared, the significance of the dust heights for the measured signal at TOA is surely dependent on the coincident water vapour and temperature profiles. Depending on the wavelength, the signal of a dust layer may be obfuscated by a particularly moist atmosphere above it: the signal of two identical dust profiles will be different if their water vapour profiles are different. Would it be possible for you to discuss, briefly or otherwise, how often the dust layers are lofted above the moistest layers of the atmosphere and to what extent this might be significant for the retrievals?

**Specific comments**

p.4, line 28: "this is a new aspect..." Could you briefly mention what the process was in the previous version, to put this into context?

p.5, line 5: this could also use a brief extra explanation, to define what the convergence criteria are.

p.7, line 22: would it be possible to quantify the number of very dusty IASI scenes which get discarded? Can this information be retained somehow in the retrieval output?

p.21, line 4: the higher dust layer is stated to be around 15th February but in the plot it

appears to be 16-18th?

p.23, line 3: does this potential underestimation of the LIDAR signal in the bottom layer of the atmosphere imply that the MAPIR retrieval has a better ability to retrieve dust at these altitudes?

Figures 11 and 12: instead of presenting the near-global maps of the AOD, it might be better here to zoom in instead on Africa and the rectangular region directly over the CATS tracks. It is quite difficult to see on the map where the pink parts of the tracks are, and to distinguish the AODs. Does the global view add any insight to this analysis compared to the regional view?

———————————————————

---

## Referee Comment (RC3) · Anonymous Referee #3 · 2 Apr 2019

The manuscript "The Mineral Aerosol Profiling from Infrared Radiances (MAPIR) algorithm: version 4.1 description and validation" by S. Callewaert et al. describes a new method applied to an algorithm that retrieves mineral aerosol profiles from infrared radiances. The new version 4.1 of MAPIR and validation results are described. As stated by the authors and proved with the validation/comparison with other products, the results show a significant increase in retrieval quality.

I have several comments that need to be addressed before this paper can be accepted for publication to AMT, most of them are for the validation/comparison part.

Main comments

[Figure]

I suggest combining the description of the lidars from Secs. 5.3.1-5.3.3 with the IASI description in Sect.2.

The algorithm is explained in detail, but I am a bit confused with the cloud screening. In Sect. 3.5 the IASI operational L2 cloud product is mentioned, while on P9 the failure of the EUMETSAT cloud filter is discussed as a possible reason for the cloud contamination. Is EUMETSAT cloud filter is used as an additional check?

Since AERONET AOD is not measuring the IR AOD (as you discuss in Sect 5.1), I would rather call the exercise as "evaluation" rather than "validation" and suggest to change the title accordingly.

AERONET coarse AOD product includes other than dust species. According-ing to Satheesh et al (https://doi.org/10.1029/2005GL024856), at high wind speeds, sea‐salt contributes 81% to the coarse mode. To classify (to some extent) the aerosol type, I suggest looking at the SSA product from the AERONET. The other useful reference for the discussion might be Khatri et al. (https://agupubs.onlinelibrary.wiley.com/doi/pdf/10.1002/2013JD019961).

Since the performance of the retrieval algorithm often depends on the aerosol load/type and surface contribution to the TOA reflectance, the evaluation results are not expected to be similar for e.g., low or high AOD loading conditions. Moreover, the number of cases with AOD>0.5 is much lower (may be up to several magnitudes lower) than the number of AOD<0.5 cases. Thus, the linear regression is not always the correct tool to evaluate the algorithm performance and I suggest to remove the regression line from Fig.6. Instead, I suggest looking at the AOD bias for certain AOD bins (e.g., Figs.1-2 in https://www.ncbi.nlm.nih.gov/pmc/articles/PMC6101972/ or Fig.3 in https://www.atmos-chem-phys.net/18/11389/2018/).

I have some doubts on how meaningful the mean bias of -0.04 is for the estimation of the MAPID AOD quality. To my rough estimation from Fig.7, only ca 10% of matches hit the bin which includes the number of -0.04; the highest probability (∼0.16) is for

slightly positive bias and the probability of the abs difference of >0.1 is high (ca >0.40). Thus, the spread is essential, even though the mean bias is low. The intercept (Table A1) is positive for most of the stations and on average is as high as ca 0.08 (my rough estimates).

The deviation of the cumulative extinction is smaller over ocean (Table1). Is that because P7L6: "we believe existing databases of ocean Ts are more reliable than land Ts" or other reasons exist? However, as for the comparison with AERONET, the results for inland stations look better than for coastal/island stations.

The transition in land/ocean AOD is not smooth (Fig 3a,b; Fig.4). Please, discuss the reason(s).

Specific comments

P2L25: OMI retrieves AOD at UV and interpolates AOD to 500nm using the Angström.

P3L15: I suggest the following changes: …. updated algorithm is presented; AOD product is evaluated against AERONET and compared.…. . The work is organized as follows.

P13L10: Please, specify the wavelengths here.

Fig.6, second column. Here I suggest to color the dots with AERONET SSA, when retrieved and add the corresponding discussion on the sea salt contribution to the AERONET coarse AOD.

Fig.7 Please, add the grid.

P16L32 One period, 1-12 July 2010, is mentioned twice

Table 1. I suggest making the case specs (e.g., CALIOP, All data) more visible by adding additional horizontal line below the case name, or/and changing the fonts and/or moving the name to the left.

P22L6: I suggest to specify months here

P23L3: Please, provide a reference, where the radar limit for the close to surface measurements is specified and discussed.

---

## Author Comment (AC1) · 28 May 2019

**Author comments AMT 2019-84**

**Callewaert Sieglinde et al.**

**1** Comments referee #1**

**1.1 Main comments**

The authors indicate the existence of parallel efforts such as the ones described in Clarisse et al. (2019) and by Cuesta et al. (2015) to make mineral aerosol retrieval. Although these efforts are mentioned in the introduction there is no mention to the reader of how the presented work here differ by its methods upon these two other studies. I am not asking for quantitative statements but rather that the readers be made aware of the limitations of each of the studies.

Response: Clarisse et al., 2019 provide a long-term data set of AOD and aerosol mean layer altitude retrieved from IASI measurements. This data set does not contain aerosol profiles and differs from the MAPIR data set as such. We believe this should be clear from the way it is formulated in the manuscript.

It is true that the existence of the aerosol profile product by Cuesta et al., 2015 might be confusing for the reader. To our knowledge, there is no long-term global data set available using the method described in Cuesta et al. (2015). This we see as the main added value of MAPIR compared to existing products: MAPIR provides the first global data set containing aerosol profiles over a long time period.

To make this more clear to the reader, we added the following: "The main differences between MAPIR and this alternative study are that Cuesta et al. (2015) follow an autoadaptive Tikhonov-Phillips-type approach and their method has until now only been applied to a very limited number of dust events, while MAPIR provides a global data set over a long time period using optimal estimation."

A lot of effort is devoted to improvements in the algorithm so that the screening of the scene is improved and the detectability of dust over desert areas with low emissivity becomes feasible. I was surprised that the dust properties chosen in the retrieval were not looked at critically. With regards to either the particle size distribution, and the refractive indices in the shortwave taken from Volz (1972, 1973) and Shettle and Fenn (1979) there are many studies showing large deficiencies in these description of dust properties. A recent study from Ryder et al. (2018) where the full size distribution of dust is measured clearly indicates that a single mode with an effective radius of 2  $\mu$ m and a width of 2.0 accounts only for a small part of this size distribution. Have the authors tried to estimate the influence this could have on their retrievals? The following references -Dubovik et al. (2002); Sinyuk et al. (2003); Colarco et al., (2003); Balkanski et al. (2007) and Di Biagio et al. (2019)- all show that the refractive indices used here are much too absorbing compared to any dust sample examined over the last 10 years. The same question than above should be addressed by the authors: how does a correction in the refractive index use would influence their results and the description of the 3-D dust distributions that they provide.

Response: Regarding the particle size distribution, for thermal infrared retrievals it is usual to represent it by a mono-modal lognormal distribution; see for example the recent Capelle et al., 2018 (doi: 10.1016/j.rse.2017.12.008) or Clarisse et al., 2019 (doi: 10.1029/2018JD029701). The exact size distribution used in those studies and in ours are close, although not precisely identical: effective radius of 2.3  $\mu$ m for Capelle et al, of 1.66  $\mu$ m for Clarisse et al. and of 2  $\mu$ m in our work. It has previously been shown that in thermal infrared retrievals, the precise size has only a second order impact on the radiance and on the retrievals (Capelle et al. 2014 - doi:10.5194/acp-14-9343-2014; and Vandenbussche et al. 2013 doi:10.5194/amt-6-2577-2013). In the analysis mentioned by the referee (Ryder 2018 - doi: 10.5194/acp-18-17225-2018), although indeed the presence of giant mode particles is highlighted, the conclusion states that the mean effective diameter was 4  $\mu$ m, perfectly agreeing with our model. Therefore, we do believe that although not perfect the current MAPIR particle size distribution is relevant for thermal infrared (longwave) retrievals where the size sensitivity is limited.

Regarding the choice of refractive index, indeed it was shown in multiple papers, including those mentioned by the referee, that the refractive index from Volz is too absorbing in the shortwave part of the spectrum. However our retrievals are performed in the longwave. In our retrieval, we do not use exactly the Volz refractive index but the "dust-like" refractive index from the GEISA-HITRAN database, which is based on measurements by Volz but also by Shettle and Fenn. That refractive index is actually less absorbing than most of the more recent measurements in the longwave (including for example Di Biagio 2017 - doi: 10.5194/acp-17-1901-2017). That difference is minor around 900 and 1200 cm-1 (in two of our retrieval windows) but the GEISA-HITRAN is more scattering and less absorbing at about 1100 cm-1 (our third retrieval window). In Vandenbussche et al, 2013 (doi: 10.5194/amt-6-2577-2013), the first publication of this algorithm, we discussed the selection of refractive index, and how we picked the one leading to the more plausible and best quality results (in terms of RMS of spectral residuals after the retrieval, convergence rate, ...).

We agree that our current selection of refractive index and particle size is not the perfect model for the real dust (especially as we use it as globally constant), but the major steps in the algorithm development done here were the change of radiative transfer code and the adaptation of the retrieval in the logarithmic space with the modified optimal estimation method including the Levenberg-Marquardt regularisation. As we wanted to show how those modifications improved the retrievals, and because the refractive index (and particle size) are second-order parameters for thermal infrared retrievals, we decided to keep them as in the previous MAPIR versions. At the end of the manuscript, it was mentioned that "Future work to further improve the MAPIR algorithm can include the better characterization of aerosols by implementing a more complex particle size distribution and varying refractive index", highlighting that we are aware there is room for improvement. For particle size, one possibility is to use a bimodal distribution, but that also requires knowing the ratio between both modes, meaning an additional parameter to retrieve, making the retrieval even more challenging. For the refractive index, we might consider a varying refractive index, for example based on the work of Di Biagio (2017). That however requires a significant additional work linked to the selection for each scene of the best refractive index to use in the retrieval. It is therefore out of the scope of this work at its current stage.

The future work sentence in the manuscript was updated to "Future work to further improve the MAPIR algorithm can include the better characterization of the dust aerosols. Possible improvements are the use of more recent refractive index data, for example those of Di Biagio et al. (2017) and the use of a bi-modal particle size distribution. Both represent significant scientific work: the development of an automated selection of the best refractive index and/or the retrieval of an additional parameter being the ratio between the 2 modes of the particle size distribution."

When examining the AERONET coarse mode AOD, I did not see any discussion about the possible influence of seasalt on these AOD. This would be particularly sensitive for marine or coastal sites near sea-level and could explain a good part of the discrepancies at these sites. Please indicate it, or try to estimate how much the total coarse mode AOD could deviate from the dust coarse mode AOD at these sites.

Response: Indeed sea salt can contribute significantly to the AERONET coarse mode AOD, especially when the dust load is very limited or absent. Therefore, the presence of sea salt would indeed impact the comparisons between IASI MAPIR dust AOD and AERONET coarse mode AOD. In particular it could impact the correlation at coastal low altitudes stations in places where dust is not present along the whole year. For example, we think it is the case in Guadeloupe (Figure 1 in the appendix of this document) where the AERONET coarse mode AOD is constantly higher than the MAPIR dust AOD. In the winter especially, it is highly improbable that the almost constant non-negligible AERONET coarse mode AOD in Guadeloupe is dust.

In the new section 2.2 describing the AERONET product, we added the sentence: There is currently no aerosol type specification in the AERONET product, and the coarse mode mainly contains mineral dust, sea salt and/or volcanic ash.

In section 5.1 discussing the AOD comparisons, the following changes were made:

(1) A last requirement is that the AERONET station should be dusty enough: only sites for which there is a sufficient amount of dust measured are included, the median of the AERONET coarse mode AOD at 500 nm over the considered time period should be higher than 0.05. [...] was replaced by Finally, we considered only sites for which the median of the AERONET coarse mode AOD at 500 nm over the considered time period is higher than 0.05. [...] was replaced by Finally, we considered only sites for which the median of the AERONET coarse mode AOD at 500 nm over the considered time period is higher than 0.05. As mentioned in Sect. 2.2, the coarse mode AOD contains all coarse mode aerosols, i.e. mainly dust, sea salt and volcanic ash. The selection therefore does not ensure the presence of only dust at those selected AERONET sites. [...].

(2) After the paragraph describing in general the AOD correlation between MAPIR and AERONET, we added: The coastal stations where the presence of sea salt aerosols plausibly impacts the AERONET coarse mode AOD and its correlation with MAPIR dust AOD are indicated with an asterisk in Table A1.

(3) In the discussion about transport to the Caribbean, we added: However, those are coastal stations where the coarse mode probably contains sea salt aerosols with a possible impact on the AERONET coarse mode AOD.

(4) Three of the sites with a weak or very weak correlation are situated in the American continent: Arica, Bakersfield and UPC-GEAB-Valledupar. The reason for this discrepancy is not clear. They are situated in regions that are not known for the presence of dust. For UPC-GEAB-Valledupar and Arica, the AOD values from AERONET are higher than the MAPIR AOD. This could indicate that there are other coarse aerosol types measured with AERONET, to which MAPIR is not sensitive. was replaced by Three of the sites with a weak or very weak correlation are situated in the American continent, in areas not known for the presence of dust: Arica, Bakersfield and UPC-GEAB-Valledupar. Arica is a coastal station, potentially experiencing sea salt aerosols. For the other 2 stations, the reason for the discrepancy is not clear.

I propose a change in the structure of the text of paragraph 5.3. The description of the Lidar characteristics for the 3 lidars at MBour, at Al Dhaid and on-board the space station should have been given on an earlier part of the paper so that the authors focus only on the comparison which is the title of paragraph 5.3. This whole paragraph needs to be better organized and better written if you want to keep the attention of the reader. This paper is relatively long so this part has to be well written.

Response: The description of the three lidar instruments is moved to a new Section 2 called 'Instruments'. To be consistent along the manuscript, we also added a short description of AERONET and CALIOP in this new section. It should now contain information on all instruments used in this study.

And last but not least of the major comments, that Data availability statement at the bottom of page 28 is very fuzzy as it is. Since this project is financed by Copernicus, the data availability is mandatory and cannot be delayed in time. How can someone interested in studying this dataset access it?

Response: We agree that the sentence could lead to some confusion. Part of the data was submitted to Copernicus: the dust AOD (550 nm and 10  $\mu$ m) and the dust mean altitude. The full profiles were not submitted because they can not be accommodated by the Copernicus system at the moment. The Copernicus funding was only to deliver AOD and if possible mean altitude, but the fact that MAPIR provides profiles does not matter here. We are working on a way to make those profiles available through our institute but it takes time to do it properly. In the meanwhile, interested scientists are encouraged to contact the authors to obtain profile data. In addition, there is a delay on the side of Copernicus to make the delivered data (AOD and mean altitude) available, which is out of our hands and is delicate to mention in a publication. Again, in the meanwhile, interested scientists may contact us for the data.

The sentence in the manuscript was modified as follows: "Under the Copernicus Climate Change Service aerosols project, the MAPIR dust 10  $\mu$ m and 550 nm AOD and the MAPIR dust aerosol mean altitude were submitted to the Copernicus Climate Data Store where they currently undergo technical processing. The full profiles (and the AOD and mean altitude) from MAPIR are available upon request to the authors".

**1.2** Minor Comments**

In the introduction, the authors should mention that 3-D fields of dust based upon observations are described in Ridley et al. (2016).

Response: The article suggested by the referee is about seasonal AOD estimated using a combination of several satellites, models and AERONET. We don't see why this should be mentioned in the introduction as AOD is not a 3-D field.

Page 6, line 16: please delete the sentence: Above 7 km there is rarely found mineral dust particles, as is shown by a CALIOP based 3-D climatology described in Winker et al. (2013). This statement is inaccurate, many lidar profiles above the Mediterranean Sea show dust plumes above 7 and even 10 km. Hence, saying that dust is rarely seen above 7 km can mislead readers.

Response: The sentence is deleted.

Page 16, line 3: you mention that you chose 4 surface stations to conduct a more thorough comparison with AERONET, please indicate how these stations were chosen.

Response: Those 4 AERONET sites shown in Figure 6 are more an illustration to show how the time series of AERONET and MAPIR compare than they are an additional comparison. We performed no additional study on these specific sites. The stations were chosen quite randomly, although we wanted them to have enough dusty events and with a continuous time series over a period that is long enough. We included them in the manuscript as we believe that the time series give a different view on the AOD comparison between MAPIR and AERONET that could be interesting for the reader.

To make it more clear, we changed the sentence to: "To illustrate the similarities and differences between MAPIR AOD and AERONET AOD in an alternative way, Fig. 6 shows time series for the AOD at 4 AERONET stations."

Page 16, line 18, please change: Indeed, for the a priori, a monthly climatology over 8 years is used (...) To A monthly climatology over 8 years is used for the a priori (...)

Response: Done.

Page 16, line 31-32: The first period is identical to the same time and region used by Kylling et al. (2018): 18.., please indicate the lat, lon of the region that you mention here.

Response: The coordinates of the region are mentioned in the next sentence: "These dates cover four desert dust events in the region between 0-40 N and 80 W-120 E."

**Page 16; in paragraph 5.2 you should indicate that if the dust layer is situated above 7 km, it will be missed by your algorithm.**

Response: This is not the case. Indeed, the signature of the dust will be present in the spectra even if the dust is higher than 7 km, therefore the retrieval will be performed. Obviously, if the dust layer is above the final layer from the retrieval grid, the retrieval will be biased, especially in terms of vertical distribution and mean altitude. It will result in the dust being retrieved in the last retrieval layer (6 to 7 km altitude) and the concentration/AOD being biased depending on the temperature difference between that retrieval layer and the real dust altitude. In the comparisons between IASI and CALIOP mean altitude, we think that this has no consequence. Indeed, as can be seen in the detailed Figure 5 in Kylling et al (2018), there are no comparisons where the mid layer altitude is higher than 5 to 5.5 km. The corresponding detailed plot for the current MAPIR version is not included in the manuscript, but the sample was identical for 2010 and is very similar for 2017 with also no mean altitude from CALIOP higher than 5 to 5.5 km.

Page 21 lines 19-20: stating that "This qualitative analysis of aerosol profiles at MBour supports our confidence in the value of the new MAPIR algorithm. Is not justified since we do not have, as of today, a golden standard for dust profiles to judge when we can be confident in a dust retrieving algorithm. Please delete this sentence.

Response: It is true there is no golden standard for assessing the quality of a dust profile retrieving algorithm. However, we believe that if two independent measurement systems show similar patterns it increases the reliability of these patterns. The sentence does not state that our algorithm provides true profiles in any case, it is just saying that the similarities between the two data sets confirm our confidence in that MAPIR is able to provide reasonable profiles. We would leave the sentence as it is.

Page 25 line 14, the work you do here is more an evaluation than a validation since you would need very well defined uncertainties on the dust quantities that are measured to make that validation. I propose that you change the term validation to evaluation in the title of this manuscript and change the text provide validation to evaluate in this sentence.

Response: Indeed, it is more correct to call the performed analyses an evaluation rather than a validation, as also pointed out by another referee. We have changed all occurrences of 'validation' by 'evaluation'.

**Page 27, line 12, there is a typo that your co-authors should have picked up: the units of extinction should be $km^{-1}$ and not km.**

Response: This is not true, the units don't correspond to an extinction but to the difference in cumulative extinction altitude between CALIOP and MAPIR dust layers. As explained in the text, this is the altitude where the dust cumulative extinction at 532 nm is half of the total extinction column. Since the values mentioned in the conclusion correspond to a measure of altitude, the unit is km.

Page 28, line 8, there is one comma too many, please delete the comma and change the text from: In Al Dhaid, United Arab Emirates, almost all dust events that were detected by the

lidar during the two-month comparison period, were also seen in the MAPIR data. To In Al Dhaid, United Arab Emirates, almost all dust events that were detected by the lidar during the two-month comparison period were also seen in the MAPIR data.

Response: Done.

**2** Comments referee #2**

**2.1 Main comment**

One broader question that I have, which I feel is considered implicitly but not explicitly throughout the manuscript, is whether there is much to be said about the relationship between the IASI-derived atmospheric profiles of water vapour (and temperature) and the dust aerosol profiles (validated in Sections 5.2 and 5.3). In the infrared, the significance of the dust heights for the measured signal at TOA is surely dependent on the coincident water vapour and temperature profiles. Depending on the wavelength, the signal of a dust layer may be obfuscated by a particularly moist atmosphere above it: the signal of two identical dust profiles will be different if their water vapour profiles are different. Would it be possible for you to discuss, briefly or otherwise, how often the dust layers are lofted above the moistest layers of the atmosphere and to what extent this might be significant for the retrievals?

Response: Indeed, we mention the impact of the quality of the IASI level 2 vertical profiles of temperature and humidity on the quality of our dust retrievals. However, that impact is relatively limited and indeed can be evaluated through the difference in validation for different time periods as discussed in the manuscript.

Our retrieval windows are not in the main water vapour absorption bands. They do however contain small absorption bands and the continuum effect, obviously. A strong humidity never saturates the TIR spectrum in those windows, leaving the opportunity to observe dust in all cases including particularly moist atmospheres. The change in the depth of the water vapour absorption bands in the spectral windows used in the retrieval will mostly impact the spectral residuals after the retrieval. However, the change in the continuum also affects the spectral "baseline", with an effect similar to that of the surface temperature, therefore affecting the retrieval of the latter, which itself also affects the retrieval of the aerosols. This has potentially more impact over land as our retrieval is set up with a Ts a priori standard deviation of 15% over land and 5% over ocean, making the Ts retrieval over land much more flexible. This is necessary because of the high uncertainty in the EUMETSAT IASI land Ts (in deserts mainly), especially under dusty conditions.

To test more specifically the effect of a change in relative humidity, we undertook the retrievals for the 9 June 2018 (the day used as example in the manuscript) with the relative humidity set to 90% of the value from IASI level 2 data normally used in the retrievals. This 10% change is driven by the IASI scientific objective of 10% accuracy in relative humidity, and from IASI validation results for level 2 version 5 (August et al, JQSRT 2012) showing an RMS of about 10% in relative humidity (and up to 20% for the lowest layers). The IASI level 2 version 6 has improved validation results in the lowest layers with about 10% standard deviation of comparisons with ECMWF (IASI L2 PPF 6.4 validation report, EUM/RSP/REP/18/974859). The results of our test retrievals shows that an uncertainty in water vapour at altitudes above 6 km has negligible impact on the MAPIR retrievals. A 10% change in relative humidity below 2 km, in the 2 to 4 km layers or in the 4 to 6 km layers changes the MAPIR dust mean altitude by at most 0.5 km and the TIR AOD by at most 0.06. See Figure 2 in the appendix of this document for an example. Depending on the location (and on the difference between the surface temperature and the aerosols temperature), a lower humidity can lead to a higher or lower AOD and respectively a lower or higher mean altitude. An increase of 10% in humidity has the opposite effect than the decrease. Overall it is improbable that the whole water vapour profile is shifted and more probable that there are some positive and some negative differences with respect to the true profile (indeed the water vapour validation does not show a constant shift). Therefore the impact on the dust retrievals of different water vapour biases at different altitudes should partially compensate and the total effect on dust aerosols retrievals is limited. We did not observe a significant effect of the relative altitude of the dust with respect to the altitudes where we modified the humidity.

A similar analysis was done with the temperature profile (see Figure 3 in the appendix of this document as example). We shifted the whole profile by 1 K (a layer by layer approach here has little sense as such

a strong jump in the temperature is nonphysical). This shift of the entire temperature profile leads to a maximum 0.6 km shift the MAPIR mean dust altitude and a maximum TIR AOD change of 0.09. Again such a bias of the temperature profile is improbable, therefore these test results provide an upper limit to the impact of uncertainty of the temperature profile on the MAPIR dust retrievals.

**2.2 Minor comments**

**p.4, line 28: this is a new aspect... Could you briefly mention what the process was in the previous version, to put this into context?**

Response: Indeed, this was not clear from the text. A clarification was added: "This is a new aspect with respect to previous MAPIR versions, where the ordinary Gauss-Newton iteration method was used."

**p.5, line 5: this could also use a brief extra explanation, to define what the convergence criteria are.**

Response: The convergence criteria are not mentioned in the manuscript as adding two extra equations would probably make the paragraph too heavy. We use the standard convergence criteria as mentioned in Rodgers (2000). Therefore we changed the sentence as follows: "The iterations are stopped when the steps both in state space and measurement space are small enough or after 20 steps, whereby the retrieval is signalled as unsuccessful. The convergence criteria on the step sizes is taken from Rodgers (2000, p90), with  $\epsilon = 10^{-1}$ ."

**p.7, line 22: would it be possible to quantify the number of very dusty IASI scenes which get discarded? Can this information be retained somehow in the retrieval output?**

Response: No, this is not possible. The discarded scenes due to the cloud filter are not treated any further in the retrieval process. To quantify when the center of a plume was flagged as a cloud, one would need to do a post-retrieval analysis on the pixels surrounding such a discarded scene. This would lead to a completely new study and falls outside the scope of this work.

**p.21, line 4: the higher dust layer is stated to be around 15th February but in the plot it appears to be 16-18th?**

Response: That is correct, the two co-colocated MAPIR retrievals involved actually correspond to the IASI overpass on the morning of 16 February and 17 February. We changed it to be 'around 17 February'.

**p.23, line 3: does this potential underestimation of the LIDAR signal in the bottom layer of the atmosphere imply that the MAPIR retrieval has a better ability to retrieve dust at these altitudes?**

Response: No, it doesn't imply that. It is just saying that the lidar might be underestimating the lowest layer because of instrumental features. This has no connection with the performance of MAPIR in that layer at all. The only thing it implies is that the comparison in the lowest layer is less trustworthy.

Figures 11 and 12: instead of presenting the near-global maps of the AOD, it might be better here to zoom in instead on Africa and the rectangular region directly over the CATS tracks. It is quite difficult to see on the map where the pink parts of the tracks are, and to distinguish the AODs. Does the global view add any insight to this analysis compared to the regional view?

Response: The AOD maps were replaced with regional maps over the selected part of the CATS track, for example see Figures 4 and 5 in the appendix.

**3** Comments referee #3**

**3.1 Main comments**

**I suggest combining the description of the lidars from Secs. 5.3.1-5.3.3 with the IASI description in Sect.2.**

Response: Done, as also suggested by referee #1. Section 2 is expanded with subsections of all used instrument descriptions.

The algorithm is explained in detail, but I am a bit confused with the cloud screening. In Sect. 3.5 the IASI operational L2 cloud product is mentioned, while on P9 the failure of the EUMETSAT cloud filter is discussed as a possible reason for the cloud contamination. Is EUMETSAT cloud filter is used as an additional check?

Response: These two cloud filters are the same. It is true that the different names can be confusing. To be consistent, we changed any occurrence of the cloud filter to "IASI operational level 2 cloud product".

**Since AERONET AOD is not measuring the IR AOD (as you discuss in Sect 5.1), I would rather call the exercise an evaluation rather than validation and suggest to change the title accordingly.**

Response: As also suggested by another referee, the undertaken comparison exercises in the validation section are more an evaluation. Therefore we have changed the title and other occurences of *validation* to *evaluation*.

AERONET coarse AOD product includes other than dust species. According to Satheesh et al (https://doi.org/10.1029/2005GL024856), at high wind speeds, sea salt contributes 81 % to the coarse mode. To classify (to some extent) the aerosol type, I suggest looking at the SSA product from the AERONET. The other useful reference for the discussion might be Khatri et al. (https://agupubs.onlinelibrary.wiley.com/doi/pdf/10.1002/2013JD019961).

Response: The possible impact of the sea salt aerosols was discussed in an answer to Referee 1 (third question, bottom of page 2 of this document) and changes were done to the manuscript accordingly.

An attempt to classify the AERONET aerosol type using SSA could indeed be done. However, that should be done very carefully and should represent a separate work by itself, with specific validation. Otherwise, the comparison of MAPIR dust AOD would not anymore be against a standard and validated reference data set. In addition, the AERONET inversion description documentation, mentions that: "NOTE: The fine and coarse modes of single scattering albedo are technically estimated, however, it is not advised to use these values for the physical interpretation because the retrieval is implemented under assumption that complex refractive index is the same for all particle sizes." Therefore, we really think that although the AERONET aerosol typing seems extremely interesting, it is outside the scope of this work and it requires very specific expertise which we do not have.

Since the performance of the retrieval algorithm often depends on the aerosol load/type and surface contribution to the TOA reflectance, the evaluation results are not expected to be similar for e.g., low or high AOD loading conditions. Moreover, the number of cases with AOD > 0.5 is much lower (may be up to several magnitudes lower) than the number of AOD < 0.5 cases. Thus, the linear regression is not always the correct tool to evaluate the algorithm performance and I suggest to remove the regression line from Fig. 6. Instead, I suggest looking at the AOD bias for certain AOD bins (e.g., Figs. 1-2 in www.ncbi.nlm.nih.gov/pmc/articles/PMC6101972/ or Fig. 3 in www.atmos-chem-phys.net/18/11389/2018/).

Response: New plots were made to replace the linear regression analysis in Fig.6 (See Figure 6 in appendix). Now the plots represent the distribution of the MAPIR - AERONET AOD difference in function of AERONET AOD as a scatter plot. In addition, 5 bins of equal sizes were composed of which the median and interquartile range (IQR) of the AOD differences were calculated and shown on top of the scatter plot. It shows that indeed the number of low AOD cases is much higher than the number of high AOD cases. Moreover, we see that the low AOD scenes tend to have a small positive bias while the medians of the more dusty cases are close to 0 or slightly negative.

We have added the following in the manuscript: "Another way of showing this are the plots in the second column of Fig. 6. It presents a scatter plot of the AOD differences in function of the size of the AERONET AOD. Additionally, the data per station is split into AOD bins of equal quantity. Binned medians (black dots) and interquartile ranges (IQR, vertical black lines) of the AOD differences are shown on the plots. For example at Tunis Carthage, we see in the AOD difference scatter plot that most of the observations are low AOD cases with small positive bias, 4 out of the 5 AOD bins are situated below 0.1. The AOD bin of larger AOD values shows a slightly negative bias, thus they are generally underestimated. This negative trend, positive bias for low AOD and negative bias for higher AOD, is to some extent present at the other stations in Fig. 6 too. "

As we see the added value in presenting our data this way, we added a similar plot for all AERONET data to the bias histogram in Fig. 7 (see response of the next comment).

I have some doubts on how meaningful the mean bias of -0.04 is for the estimation of the MAPIR AOD quality. To my rough estimation from Fig.7, only ca 10% of matches hit the bin which includes the number of -0.04; the highest probability ( $\sim 0.16$ ) is for slightly positive bias and the probability of the abs difference of > 0.1 is high (ca > 0.40). Thus, the spread is essential, even though the mean bias is low. The intercept (Table A1) is positive for most of the stations and on average is as high as ca 0.08 (my rough estimates).

Response: The mean bias of MAPIR AOD with respect to AERONET AOD actually is 0.04 instead of -0.04. This misinterpretation happened because the histogram in Fig.7 of the submitted manuscript was plotted with the difference AERONET - MAPIR, and the bias was calculated with the same data. It is however common to mention the mean difference as MAPIR - AERONET, which leads to a reverse sign. It has been corrected in the manuscript. We believe reporting the mean bias is more meaningful now as it is consistent with the positive intercepts at the stations.

Further, we have made the histogram plot more clear (see Fig.7b in appendix): horizontal grid lines were added, the borders of the histogram boxes were added and also the position and width of the bins were adapted. Now it is well visible that almost 50% of the absolute differences is below 0.05 and more than 70% is below 0.1, which means the MAPIR AOD quality is indeed decent. We have added the standard deviation ( $\sigma = 0.16$ ) and the following sentence in the manuscript: "... and more than 70% of the absolute differences fall below 0.1".

Additionally, we added another plot to Figure 7: a similar scatter plot as the ones replaced in Fig. 6, but now for all AERONET data (See Fig. 7a in appendix). We believe this gives some additional information to the reader about the difference distribution. It shows that most of the positive bias comes from low AERONET AOD measurements while the negative bias is mainly from the more dusty scenes. Also, it visualizes the positivity constraint on the AOD as an imaginary line in the lower left corner of the scatter plot.

Accompanying this new plot, we added the following paragraph in the manuscript: "...Figure 7(a) shows the same kind of plot but for all AERONET stations combined and split up into 10 bins of equal size. The total number of points used for these statistics are 76976, for the whole time period over the 72 selected AERONET stations. The binned medians show that the low AOD cases (AOD < 0.1) have a small positive bias, the cases with AERONET AOD between 0.1 and 0.4 have almost no bias but there is a bigger spread, and the most dusty scenes (AOD > 0.4) show a small negative bias. The imaginary line that is visible in the lower left part of the scatter plot is due to the positivity constraint on the AOD values." The deviation of the cumulative extinction is smaller over ocean (Table1). Is that because P7L6: we believe existing databases of ocean Ts are more reliable than land Ts or other reasons exist? However, as for the comparison with AERONET, the results for inland stations look better than for coastal/island stations.

Response: The explanation about the better existing databases for Ts over ocean than land justifies that in the retrieval we use a different standard deviation for the a priori value of Ts. This is probably part of the reason for a different spread of the mean altitude from MAPIR versus CALIOP over ocean and land. We think that overall this is the result of an easier case over ocean for the retrieval: the surface emissivity and temperature are less uncertain and the plume height is relatively constant (therefore with less deviation from the a priori).

We added the following sentence in section 5.2 of the manuscript: We also observe a lower standard deviation over ocean than over land. This is probably linked to the fact that retrievals over ocean are less uncertain: the surface emissivity and temperature are more stable. In addition, the plume height is more constant over ocean (no local source), therefore less deviating from the a priori.

The fact that the AERONET AOD validation shows better results inland than for coastal stations is not so clear. Some inland stations show a weak correlation and some coastal stations show a very good correlation. Part of the weak correlation at coastal stations could be due to the sea salt issue mentioned by this referee and another one.

**The transition in land/ocean AOD is not smooth (Fig 3a,b; Fig.4). Please, discuss the reason(s).**

Response: The referee probably refers to the land/ocean transition along the west African coast. The coastline AOD transition in the Arabian Peninsula in this plot is very smooth. For other days, a smooth transition is observed along the African coastline but it appears that around 9 June 2018 the plume was a bit unusual. A similar dust plume with also a discontinuity along the African coast was observed by MODIS (the data can be visualized at https://worldview.earthdata.nasa.gov). However, there is a small area near the South coast of Mauritania where the MAPIR AOD is indeed quite low while it is higher to the North and West. In that area (S Mauritania) we see an RMS that almost reaches our quality filter of 1 K. This could indicate that MAPIR is slightly underestimating the dust load in that particular case.

We have added the following sentences to the manuscript: The apparent discontinuity along the West coast of Africa (which can be better seen in Fig. 4), is probably caused by the shape of the dust plume itself. This event observed by MODIS shows similar patterns (the data can be visualized at https://worldview.earthdata.nasa.gov). However, a small area near the South coast of Mauritania where the MAPIR AOD is low compared to its surroundings, shows relatively high RMSSR values almost reaching our quality filter of 1 K. This could indicate that MAPIR is slightly underestimating the dust load in that particular area.

**3.2** Minor comments**

**P2L25: OMI retrieves AOD at UV and interpolates AOD to 500 nm using the Angstrom.**

Response: To make the interpretation of the sentence more correct, regarding the wave numbers at which the mentioned AOD products are delivered, it has been changed as follows: "... is a parameter that many sensors provide, such as.... They measure in the UV, visible or near-infrared and typically report AOD around 550 nm."

**P3L15: I suggest the following changes: ... updated algorithm is presented; AOD product is evaluated against AERONET and compared... The work is organized as follows.**

Response: In general, we have changed *validated* by *evaluated* along the manuscript because it is indeed a more accurate term, considering the comparison exercises performed.

**P13L10: Please, specify the wavelengths here.**

Response: Done. We convert to 550 nm.

Fig. 6, second column. Here I suggest to color the dots with AERONET SSA, when retrieved and add the corresponding discussion on the sea salt contribution to the AERONET coarse AOD.

Response: Not done, linked to our response concerning using AERONET SSA.

**Fig. 7 Please, add the grid.**

Response: Done. See Figure 7 in the appendix of this document.

**P16L32 One period, 1-12 July 2010, is mentioned twice**

Response: Thank you for pointing out this typo which the authors have missed. The correct fourth period (14–20 September 2010) is added.

Table 1. I suggest making the case specs (e.g., CALIOP, All data) more visible by adding additional horizontal line below the case name, or/and changing the fonts and/or moving the name to the left.

Response: The case specs were moved to the left and put in bold.

**P22L6: I suggest to specify months here**

Response: It is indeed more clear to repeat the exact months also here in the text, not only on the corresponding figure and the description of the instrument. Especially since the latter is moved to another section following a previous comment. A clarification of the considered time period is added: A comparison of the 2 months measurements (March to April 2018) [...]

**P23L3: Please, provide a reference, where the radar limit for the close to surface measurements is specified and discussed.**

Response: The detailed information can be found in Engelmann et al., 2016, which was cited on p21 line 28 (now moved to section 2.5). The reference is cited again where the lidar limit is mentioned.

**A Figures**

Figure 1: Time series of the available AERONET data at Guadeloup stations showing the possible contamination of sea salt. The blue dots are the AERONET coarse mode AOD, the red dots are the matched MAPIR AOD converted to the same wavelength.